# Towards Global Expert-Level Mixed-Precision Quantization for Mixture-of-Experts LLMs

## Abstract

Mixture-of-Experts large language models (MoE-LLMs) achieve state-of-the-art performance across diverse language tasks but incur substantial memory overhead due to their massive expert parameters. Mixed-precision quantization, which allocates different bit-widths to experts according to their importance, has emerged as a promising technique for reducing the memory consumption of MoE-LLMs. However, we identify two key limitations in existing MoE-LLMs quantization methods: (1) expert importance is estimated only locally within each MoE layer, failing to capture global importance across the model and leading to suboptimal bit-width allocation; and (2) expert quantization substantially alters the dynamics of MoE routers, yet this effect is often overlooked, resulting in suboptimal routing. In this work, we propose **G**lobal **E**xpert-level **M**ixed-precision **Q**uantization (**GEMQ**) to overcome these limitations and enable extreme low-bit quantization. First, we introduce a global expert bit-width allocation method that formulates a linear programming model based on quantization error analysis to capture global expert importance. Second, we propose an efficient global router fine-tuning approach that adapts routers to quantized experts, enabling optimal routing. Additionally, we integrate the two techniques into a progressive quantization framework that leverages the previously quantized and fine-tuned model for expert importance estimation, enabling more accurate allocation and improved performance. Extensive experiments show that our approach substantially reduces memory usage and improves inference speed while incurring minimal performance degradation.

## 1 Introduction

Large Language Models (LLMs) have achieved state-of-the-art performance on a wide range of natural language processing tasks (Minaee et al., 2024). Among recent advancements, Mixture-of-Experts (MoE) architectures have emerged as a prominent scaling paradigm (Shazeer et al., 2017; Muennighoff et al., 2024; Jiang et al., 2024; Dai et al., 2024; Yang et al., 2025). MoE models utilize a sparse architecture wherein a routing mechanism selectively activates a small subset of expert networks for each input. This design facilitates efficient scaling and enables MoE-LLMs to match the performance of dense counterparts with a fraction of the computational cost. However, while this approach reduces the computational load per input, the total number of model parameters remains unchanged. All experts must be co-located in memory during inference, resulting in a substantial memory footprint that presents a significant deployment challenge (Kim et al., 2023b; Li et al., 2024; Huang et al., 2024a). Even high-end GPUs such as the NVIDIA H100-80GB are insufficient to accommodate typical MoE models like Mixtral-8×7B (Jiang et al., 2024) in full-precision. Therefore, effective model compression is critical for practical deployment of MoE-LLMs.

Quantization, which reduces the numerical precision of model parameters, has emerged as a promising compression technique widely applied to dense LLMs (Dettmers et al., 2022; Frantar et al., 2022; Lin et al., 2024). However, naively applying these methods to MoE-LLMs overlooks the unique properties of MoE architectures and leads to severe performance degradation (Li et al., 2024). On one hand, the primary objective of MoE-LLMs quantization is to reduce the size of expert parameters, which typically account for over 90% of the total model parameters and dominate memory consumption (Kim et al., 2023b; Huang et al., 2024a). On the other hand, recent studies (Lu et al., 2024; Huang et al., 2024a) have shown that experts exhibit distinct activation patterns, indicating

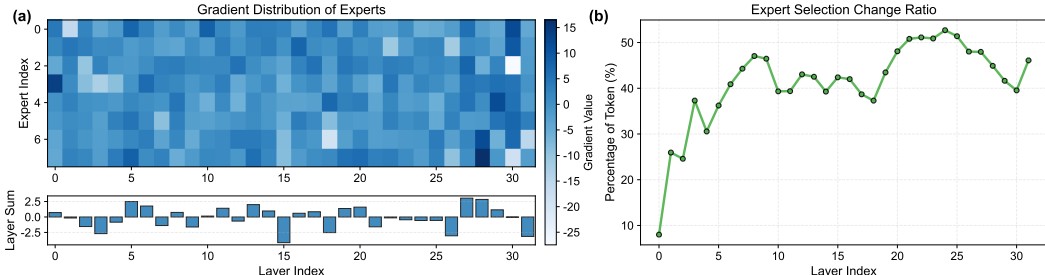

Figure 1: **Motivations.** (a) The gradients of expert weights vary not only within a layer but also across layers, indicating that different layers exhibit varying levels of importance; and (b) over 40% of tokens are routed to different experts after 1.5-bit quantization, revealing substantial distortion in router distributions. Statistics are computed from the Mixtral-8×7B model on WikiText2.

that not all experts are equally important. This motivates expert-level mixed-precision quantization for MoE-LLMs (Li et al., 2024; Huang et al., 2024a), where experts are assigned different bit-widths according to their relative importance. Specifically, expert bit allocation is formulated as a linear programming (LP) problem, with statistics (*e.g.*, expert activation frequency) measured on the full-precision (FP) model serving as proxies for importance. Once the LP determines the bit assignment, a post-training quantization algorithm such as GPTQ (Frantar et al., 2022) is applied to each expert accordingly. This mixed-precision strategy substantially outperforms uniform allocation, preserving high performance even under extremely low-bit settings (Huang et al., 2024a).

Despite their effectiveness, we identify two primary limitations in existing mixed-precision quantization methods for MoE-LLMs: **(1)** bit allocation is performed locally at the layer level, considering only the relative importance of experts within each layer and assigning an identical bit budget to every layer. However, as shown in Fig. 1 (a), we observe that different layers exhibit varying levels of importance and thus necessitate different bit budgets; and **(2)** as shown in Fig. 1 (b), low-bit quantization substantially alters the dynamics of MoE routers, leading to large shifts in token-to-expert assignments. However, existing work largely overlooks this discrepancy, resulting in suboptimal routing and degraded performance in quantized models.

To address these issues, we propose Global Expert-level Mixed-precision Quantization (GEMQ), which enables effective quantization for MoE-LLMs under extreme low-bit settings. Our method addresses the aforementioned issues in both the pre-quantization and post-quantization phases. In pre-quantization, we formulate a linear programming model for expert-level bit-width allocation from an error-analysis perspective, where each expert's importance is derived globally based on the task loss. This formulation allows simultaneous consideration of expert importance across the model and naturally extends the layer-wise local LP (Huang et al., 2024a) into a global LP. In post-quantization, we propose a simple yet effective approach that adapts routers to quantized experts by fine-tuning router weights, which constitute less than 0.04% of the total parameters. This parameter-efficient fine-tuning (PEFT) strategy requires only a small calibration set and induces less than 5% time overhead in the quantization process, while enabling optimal expert selection and substantially improving quantization performance. Additionally, we analyze the loss landscape of MoE-LLMs and propose integrating the two techniques into a progressive quantization framework to alleviate inaccurate expert importance estimation under large quantization errors. Specifically, instead of using the FP model for bit allocation, we leverage the quantized and fine-tuned model under the previous bit budget, thereby mitigating the FP-quantized discrepancy and yielding more accurate expert bit-width allocation.

We conduct extensive experiments on diverse MoE-LLMs and benchmarks, demonstrating that the proposed GEMQ approach substantially reduces memory usage while largely preserving model capabilities. Specifically, when quantizing the 16-bit Mixtral-8×7B (Jiang et al., 2024) to 2.5 bits per expert, our method reduces the model size from 87 GB to 16 GB (an 82% reduction), with only a 7% accuracy drop on 5-shot MMLU (Hendrycks et al., 2020), outperforming the state-of-the-art mixed-precision quantization method PMQ (Huang et al., 2024a) by about 3%.

## 2 RELATED WORK

**Quantization for LLMs.** Post-training quantization (PTQ) has emerged as a widely used compression method for dense LLMs as it can be applied without requiring additional training (Dettmers et al.,

2022; Frantar et al., 2022; Xiao et al., 2023; Shao et al., 2023; Kim et al., 2023a; Lin et al., 2024). Quantization-aware training (QAT) is more effective in preserving the performance of quantized lightweight LLMs, though it requires substantial retraining resources (Liu et al., 2023; Chen et al., 2024). Our method integrates PTQ with parameter-efficient fine-tuning (PEFT), achieving improved performance while incurring minimal overhead. Since the primary bottleneck for generative inference with LLMs is memory bandwidth (Kim et al., 2023a; Lin et al., 2024), many studies focus on weight-only quantization (Frantar et al., 2022; Xiao et al., 2023; Kim et al., 2023a; Lin et al., 2024), where model weights are quantized to low bits while computations are performed in full precision. In this work, we also focus on weight-only quantization, as the deployment challenges of MoE models stem primarily from substantial memory pressure. On the other hand, prior work has explored the varying sensitivity of weights in LLMs and proposed mixed-precision quantization methods that allocate different bit widths accordingly (Dong et al., 2020; Li et al., 2021; Dettmers et al., 2022; 2023; Huang et al., 2024c; Ding et al., 2023; Huang et al., 2024b). However, applying these techniques to MoE-LLMs remains challenging due to the unique expert routing mechanism in MoE architectures.

**Quantization for Mixture-of-Experts LLMs.** Early studies (Kim et al., 2023b; Frantar & Alistarh, 2023) on MoE-LLMs quantization assign a uniform bit-width to all experts and directly apply PTQ methods (Frantar et al., 2022; Lin et al., 2024) developed for dense LLMs for quantization. However, such approaches overlook the sparsity inherent to the MoE architecture, leading to suboptimal performance. Li et al. (2024) pioneer the study of expert-level mixed-precision quantization, proposing a bit allocation strategy based on expert activation frequency. PMQ (Huang et al., 2024a) further advances this line of work by formulating expert bit-width allocation as a linear programming (LP) problem and employing a heuristic coefficient that combines expert activation frequency with weight statistics to determine expert importance. Building upon this, Duanmu et al. (2025) explore finer-grained sub-expert bit allocation and incorporate hardware-aware co-design into the LP formulation. However, existing approaches remain constrained to local, layer-wise bit allocation and thus fail to capture variations in expert importance across layers. Beyond mixed-precision allocation, other efforts address different aspects of MoE-LLMs quantization. Hu et al. (2025) examine the role of calibration data and propose generating an expert-balanced dataset through a self-sampling strategy. Zheng et al. (2025) also study calibration set construction by modeling the joint distribution of multiple datasets. Recent work (Chen et al., 2025; Fu et al., 2025) investigates router distribution shifts caused by expert quantization in uniform MoE-LLM quantization and proposes a layer-by-layer calibration method. However, their approach rigidly enforces alignment with the full-precision router distribution, which limits adaptability and proves suboptimal in our experiments for mixed-precision MoE-LLMs quantization.

## 3 PRELIMINARIES

**Mixture-of-Experts.** MoE-LLMs replace conventional feed-forward networks (FFNs) with MoE blocks, each comprising a router FFN and $N$ expert FFNs (Gale et al., 2023). For each input token $\mathbf{x}$, the router first computes routing logits $\mathbf{r} = \{r_0, r_1, \ldots, r_{N-1}\}$ and routing scores $\mathbf{s} = \mathrm{Softmax}(\mathbf{r})$. The top-$K$ ($K \ll N$) experts with the highest scores are then selected, and their outputs are aggregated through a weighted sum to produce the output $\mathbf{z}$ of the MoE block:

$$\mathbf{z} = \sum_{i=0}^{K-1} \frac{\mathbf{s}_i}{\sum_{j=0}^{K-1} \mathbf{s}_j} \mathrm{E}_i(\mathbf{x}), \tag{1}$$

where $\mathrm{E}_i$ denotes the feed-forward operator of the $i$-th expert FFN.

**LLM Quantization.** Quantization maps floating-point weights from the range $[\mathbf{w}_{\min}, \mathbf{w}_{\max}]$ to integers in $[0, 1, \ldots, 2^b - 1]$, where $b$ denotes the target bit-width. For LLM quantization, the primary objective is to quantize the FFNs, which constitute the majority of the model's weights. GPTQ (Frantar et al., 2022) is one of the most widely used methods for quantizing FFN weights in dense LLMs. It determines the optimal quantization mapping for each linear module by minimizing the following reconstruction error on a small calibration dataset:

$$\arg\min_{\hat{\mathbf{w}}} \|\mathbf{w}\mathbf{x} - \hat{\mathbf{w}}\mathbf{x}\|_2^2, \tag{2}$$

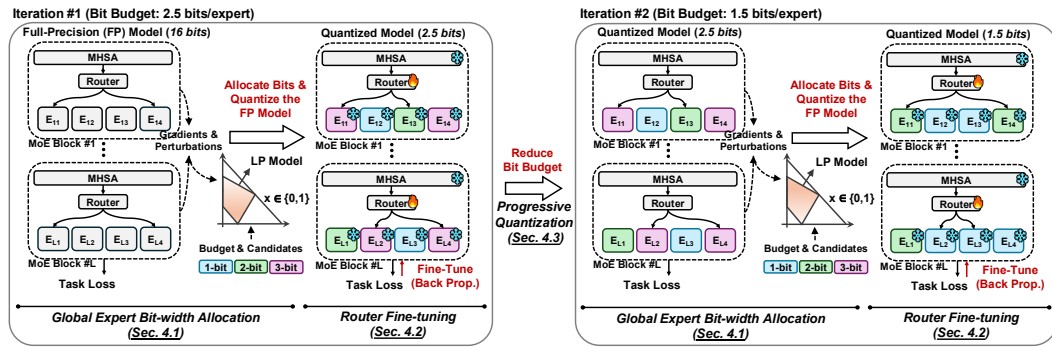

Figure 2: Overview of the proposed GEMQ framework for MoE-LLMs quantization.

where $\hat{\mathbf{w}}$ denotes the quantized weights of the linear module. Leveraging Hessian-based estimation ($\mathbf{H} = 2\mathbf{x}\mathbf{x}^\top$) and lazy batch updates, GPTQ can efficiently quantize Mixtral-8×7B (47B parameters) in about 40 minutes on a single NVIDIA H100 GPU. In this work, we focus on developing optimal mixed-precision quantization strategies for MoE-LLMs, employing GPTQ as the underlying quantization method. We note that our approach is orthogonal to other quantization techniques (Shao et al., 2023; Lin et al., 2024; Tseng et al., 2024; Ashkboos et al., 2024).

**Quantization Error Analysis.** Denote a block with weights $\mathbf{w}$, where its outputs are given by $\mathbf{z} = \mathcal{F}_\mathbf{w}(\mathbf{x})$, and $\mathcal{F}_\mathbf{w}$ represents the feed-forward function of the block. Quantization can be viewed as weight perturbation that introduces an error $\Delta\mathbf{w}$ to the original weights. To quantitatively analyze the resulting loss degradation, Nagel et al. (2020) employ a Taylor series expansion to approximate the loss increase as:

$$\mathbb{E}[\mathcal{L}(\mathbf{w} + \Delta\mathbf{w}) - \mathcal{L}(\mathbf{w})] \approx \Delta\mathbf{w}^\top \mathbf{g}^{(\mathbf{w})} + \tfrac{1}{2}\Delta\mathbf{w}^\top \mathbf{H}^{(\mathbf{w})}\Delta\mathbf{w}, \tag{3}$$

where $\mathbf{g}^{(\mathbf{w})} = \mathbb{E}[\nabla_\mathbf{w}\mathcal{L}]$ and $\mathbf{H}^{(\mathbf{w})} = \mathbb{E}[\nabla_\mathbf{w}^2\mathcal{L}]$ denote the gradient and Hessian, respectively. Assuming the pre-trained model has converged to a local minimum, the gradient term in Eq. 3 is nearly zero and can be neglected, and Hessian $\mathbf{H}^{(\mathbf{w})}$ can be approximated as the Gauss-Newton matrix $\mathbf{G}^{(\mathbf{w})} = \mathbf{J}_{(\mathbf{z})}(\mathbf{w})^\top \mathbf{H}^{(\mathbf{z})}\mathbf{J}_{(\mathbf{z})}(\mathbf{w})$ (Li et al., 2021). Due to the massive parameters in LLMs, explicitly computing the Jacobian $\mathbf{J}_{(\mathbf{z})}(\mathbf{w})$ is infeasible. Li et al. (2021) therefore propose to transform it to the perturbation of block outputs $\Delta\mathbf{z} \approx \mathbf{J}_{(\mathbf{z})}(\Delta\mathbf{w})\Delta\mathbf{w}$. Eq. 3 then becomes:

$$\mathbb{E}[\mathcal{L}(\mathbf{w} + \Delta\mathbf{w}) - \mathcal{L}(\mathbf{w})] \approx \tfrac{1}{2}\Delta\mathbf{z}^\top \mathbf{H}^{(\mathbf{z})}\Delta\mathbf{z}, \tag{4}$$

where $\Delta\mathbf{z} = \hat{\mathbf{z}} - \mathbf{z}$, and $\hat{\mathbf{z}}$ denotes the block outputs perturbed by the quantization error $\Delta\mathbf{w}$.

Given the high cost of computing the Hessian matrix $\mathbf{H}^{(\mathbf{z})}$, it is commonly approximated by the diagonal Fisher Information Matrix (FIM) (Li et al., 2021; Kim et al., 2023a) as:

$$\mathbf{H}^{(\mathbf{z})} \approx \mathrm{diag}\left(\left(\mathbf{g}^{(\mathbf{z})}\right)^2\right), \tag{5}$$

where $(\mathbf{g}^{(\mathbf{z})})^2 = \left((\partial\mathcal{L}/\partial\mathbf{z}_1)^2, \ldots, (\partial\mathcal{L}/\partial\mathbf{z}_a)^2\right)$ denotes element-wise square of gradients of $\mathbf{z}$.

# 4 METHOD

Fig. 2 presents an overview of the proposed GEMQ framework for MoE-LLMs quantization, which operates progressively. In each iteration under a given bit budget, we first apply a global allocation method to determine the optimal bit-width for each expert (Sec. 4.1). We then introduce an efficient global router fine-tuning method to solve the suboptimal router selection problem caused by expert quantization (Sec. 4.2). Finally, in Sec. 4.3, we integrate these two techniques into a progressive quantization framework to improve the estimation of expert importance under low-bit settings.

## 4.1 GLOBAL EXPERT BIT-WIDTH ALLOCATION

Given a set of candidate bit-widths $\mathcal{B}$, for example $\mathcal{B} = \{1, 2, 3\}$, and a total bit budget $B$, the objective of expert bit-width allocation is to assign a bit-width to each expert $\mathrm{E}_i$ such that the sum

of all assigned bit-widths does not exceed $B$, while minimizing the increase in task loss $\mathcal{L}_{\text{CE}}$ (*i.e.*, cross-entropy for language modeling) introduced by quantization.

To this end, we apply the approximation in Eq. 4 at the expert level, using the FIM approximation in place of the output Hessian to quantitatively evaluate how quantizing an individual expert affects task loss. Specifically, the increase in task loss caused by quantizing the $i$-th expert to $j$ bits, $j \in \mathcal{B}$, can be expressed as:

$$\mathbb{E}[\mathcal{L}_{\text{CE}}(\mathbf{w}_i + \Delta\mathbf{w}_{ij}) - \mathcal{L}_{\text{CE}}(\mathbf{w}_i)] \approx \Delta\mathbf{z}_{ij}^\top \operatorname{diag}\left(\left(\mathbf{g}_i^{(\mathbf{z})}\right)^2\right) \Delta\mathbf{z}_{ij}, \tag{6}$$

where $\Delta\mathbf{z}_{ij} = \hat{\mathbf{z}}_{ij} - \mathbf{z}_i$ denotes the perturbation of outputs in the corresponding MoE layer caused by quantization, and $\mathbf{g}_i^{(\mathbf{z})}$ represents the gradients of the corresponding layer outputs. Intuitively, this formulation estimates the increase in task loss as the effect of local layer perturbations, weighted by the squared gradients of the corresponding layer. Note that Eq. 6 is defined with respect to the task loss across all experts, making the terms directly comparable across experts, in contrast to the layer-wise reconstruction used in prior work (Huang et al., 2024a; Duanmu et al., 2025).

With Eq. 6, we obtain a measure of expert importance for bit allocation based on sensitivity to task loss: if quantizing an expert leads to a substantial increase in loss, that expert is considered important and should be allocated a higher bit-width. To determine the optimal bit-width for each experts based on the expert importance, we formulate the following binary linear programming (LP) problem, which jointly considers all experts in the model and can be solved within seconds:

$$\min_{\{x_{ij}\}} \quad \sum_{i \in \mathcal{E}} \sum_{j \in \mathcal{B}} \Delta\mathbf{z}_{ij}^\top \operatorname{diag}\left(\left(\mathbf{g}_i^{(\mathbf{z})}\right)^2\right) \Delta\mathbf{z}_{ij} \cdot x_{ij}$$

$$\text{s.t.} \quad x_{ij} \in \{0, 1\}, \ \forall i, j, \quad \sum_{i,j} j \cdot x_{ij} \leq B, \quad \sum_j x_{ij} = 1, \ \forall i, \tag{7}$$

where $x_{ij}$ is a binary variable indicating whether the $i$-th expert is assigned a $j$-bit quantization, and $\mathcal{E}$ denotes the set of all experts in the model. Inspired by Huang et al. (2024a), we enforce each layer to include at least one higher-bit expert (*i.e.*, 2- and 3-bit), which we empirically find to act as a mild regularization that alleviates inaccurate importance estimation and prevents information bottlenecks in the low-bit budget regime. Unlike existing methods that rely on heuristic coefficients with hand-tuned hyperparameters for estimating expert importance, our formulation is hyperparameter-free and readily adaptable to different MoE-LLMs. After determining the optimal bit-width assignment for each expert in the model, we apply the GPTQ (Frantar et al., 2022) algorithm for quantization.

## 4.2 Global Router Fine-tuning

Due to the unique gating mechanism of MoE, quantizing experts can substantially affect router dynamics, altering both the router's input distribution and the behavior of routed experts, which in turn changes token-to-expert assignments, as shown in Fig. 1 (b). Prior work on MoE-LLMs quantization overlook this issue, leading to suboptimal expert selection and performance degradation after quantization. To address this issue, we propose calibrating routers holistically by jointly tuning all router parameters after expert quantization to adjust the global dynamics. As shown in Fig. 2, we initialize the model with quantized weights (dequantized to floating-points in practice), freeze the attention and expert weights, and update only the router weights to minimize the cross-entropy task loss on a calibration set. The training process then guides the routers toward optimal routing decisions for expert selection. As shown in Tab. 6, since routers in MoE-LLMs typically constitute less than 0.04% of the total model parameters, the fine-tuning is inherently parameter-efficient and can be performed with minimal resources. For instance, fine-tuning the Mixtral-8×7B model on a calibration set of 128 sequences (2048 tokens each) for one epoch takes less than two minutes on three NVIDIA H100 GPUs, accounting for only 4.8% of the GPTQ quantization time. Yet this simple fine-tuning yields significant performance gains for quantized models, particularly in low-bit scenarios where router distributions change substantially, as evidenced in Sec. 5.

## 4.3 Progressive Quantization

In previous sections, we assume a valid Taylor series expansion of Eq. 3 for loss approximation when estimating expert importance. However, due to the unique loss landscape of MoE models, this

---

**Algorithm 1:** Progressive Quantization

---

**Input:** Pretrained FP model $\mathbf{Q}_0$; Calibration datasets; Target bit budgets $\{B_1, \dots, B_K\}$

Initialize an intermediate quantized model $\mathbf{Q}'$ from FP model $\mathbf{Q}' \leftarrow \mathbf{Q}_0$;

**for** *all $k = 1, 2, \dots, K$-th target bit budget* **do**

    Collect perturbations $\Delta \mathbf{z}_{ij}$ and layer output gradients $\mathbf{g}_i^{(\mathbf{z})}$ in Eq. 6 from $\mathbf{Q}'$;

    Solve the LP model in Eq. 7 under budget $B_k$ to determine expert bit-width allocation;

    Apply GPTQ pseudo quantization to $\mathbf{Q}_0$ according to the allocation results to obtain $\hat{\mathbf{Q}}_k$;

    Fine-tune routers of the quantized model $\hat{\mathbf{Q}}_k$ to obtain model $\mathbf{Q}_k$;

    Update the intermediate model for progressive quantization $\mathbf{Q}' \leftarrow \mathbf{Q}_k$;

    **if** *Router Quantization is triggered* **then**

        Quantize all routers of $\mathbf{Q}_k$ with GPTQ to the target bit-width;

**return** Quantized models $\{\mathbf{Q}_1, \dots, \mathbf{Q}_K\}$

---

approximation can become unreliable under low-bit quantization, where the quantization error $\Delta \mathbf{w}$ is large. In Fig. 3, we provide a 1-D illustration of the loss landscape of an expert in MoE-LLMs. As shown, perturbations to the weight $\mathbf{w}$ can cause abrupt changes in loss when they alter subsequent router selections. As shown in case (b), for large $\Delta \mathbf{w}$, the Taylor approximation around full-precision (FP) weights becomes inaccurate for estimating the loss of target weights $\hat{\mathbf{w}}$, due to abrupt loss changes induced by shifts in subsequent router selection. On the other hand, using weights closer to the target for estimation can alleviate the inaccuracy caused by abrupt changes, as shown in case (c). We hypothesize that a quantized model with a bit budget close to the target model can serve as such a good estimator. Nevertheless, using a quantized model remains problematic, as the quantized expert is no longer optimal and thus violates the local minimum assumption, rendering Eq. 4 inaccurate. As shown in case (d), using a fine-tuned model resolves this issue, as router fine-tuning ensures optimal expert selection and brings the quantized weights closer to the local minimum.

Based on the analysis, we propose a progressive quantization strategy that leverages a fine-tuned model with a close bit budget to improve expert importance estimation accuracy. Algorithm 1 outlines the proposed method. Specifically, let $K$ denote the number of target bit budgets, sorted in *descending order* within $[B_1, B_K]$. We begin by quantizing the model at the highest bit budget $B_1$, where the FP model is used to extract the LP coefficients in Eq. 6, followed by router fine-tuning to obtain the quantized model $\mathbf{Q}_{B_1}$. Since the quantization error $\Delta \mathbf{w}$ remains small at high bit-widths, the estimation in Eq. 6 is reliable and $\mathbf{Q}_{B_1}$ achieves strong performance. We then progressively reduce the bit budget. For the $k$-th budget ($k > 1$), instead of reusing the FP model to compute Eq. 6, we leverage the previously fine-tuned model $\mathbf{Q}_{B_{k-1}}$. This design yields more accurate expert importance estimation in low-bit scenarios and improves quantization performance.

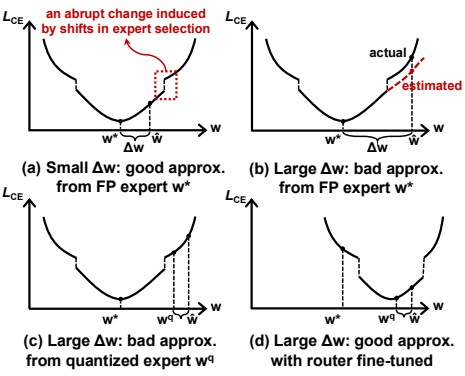

(a) Small $\Delta \mathbf{w}$: good approx. from FP expert $\mathbf{w}^\star$
(b) Large $\Delta \mathbf{w}$: bad approx. from FP expert $\mathbf{w}^\star$
(c) Large $\Delta \mathbf{w}$: bad approx. from quantized expert $\mathbf{w}^q$
(d) Large $\Delta \mathbf{w}$: good approx. with router fine-tuned

Figure 3: Illustration of different cases of expert importance estimation for target weights $\hat{\mathbf{w}}$. $\mathbf{w}^\star$ denotes full-precision expert weights.

## 5 EXPERIMENTS

**Models and Datasets.** We evaluate our method on four MoE-LLMs with varying characteristics: DeepseekV2-Lite (Dai et al., 2024), Qwen1.5-MoE-A2.7B (Yang et al., 2025), Qwen3-30B-A3B (Yang et al., 2025), and Mixtral-8×7B (Jiang et al., 2024). For evaluation, we report perplexity (PPL) on the general-domain datasets WikiText2 (Merity et al., 2016) and C4 (Raffel et al., 2020) to measure token prediction capability. To further evaluate the knowledge of the quantized LLMs, we report zero-shot accuracy on seven benchmarks using the EleutherAI LM Harness (Gao et al., 2024). Further details of the models, datasets, and evaluation protocol are provided in Appendix A.1.

Table 1: Comparison of perplexity↓ on the WikiText2 (WT) and C4 test sets, and accuracy↑ (%) across seven zero-shot tasks: PIQA (PQ), ARC-Easy (AE), ARC-Challenge (AC), HellaSwag (HS), WinoGrande (WG), MathQA (MQ), and MMLU (MM). The average accuracy (Avg.) over zero-shot tasks is also reported. #Bits denotes the average bit-width per expert. Model parameters are reported as (#Total - A#Activated). The best results in each comparison are highlighted in **bold**.

| Model | #Bits | Method | WT↓ | C4↓ | PQ↑ | AE↑ | AC↑ | HS↑ | WG↑ | MQ↑ | MM↑ | Avg.↑ |
|---|---|---|---|---|---|---|---|---|---|---|---|---|
| DeepSeekV2-Lite (15.7B - A2.4B) | 16.0 | FP | 6.31 | 9.32 | 80.09 | 76.47 | 48.98 | 78.01 | 71.51 | 39.73 | 54.90 | 64.24 |
| | 2.5 | Uniform | 8.14 | 13.59 | 76.66 | 71.34 | 45.22 | 72.44 | 69.46 | 34.77 | **48.78** | 59.81 |
| | | PMQ | 6.95 | 10.74 | 77.86 | 71.68 | 44.62 | 72.76 | 69.06 | **36.18** | 45.01 | 59.60 |
| | | GEMQ | **6.65** | **10.51** | **77.91** | **74.41** | **46.93** | **73.39** | 69.46 | 35.61 | 47.37 | **60.73** |
| | 2.0 | Uniform | 9.57 | 16.90 | 73.07 | 60.61 | 35.84 | 62.87 | 62.04 | 28.74 | 27.25 | 50.06 |
| | | PMQ | 7.99 | 12.55 | 75.14 | 61.20 | 37.37 | 66.20 | **68.11** | **30.55** | **29.06** | 52.52 |
| | | GEMQ | **7.27** | **11.93** | **76.55** | **68.10** | **40.53** | **68.43** | 66.38 | 27.97 | 28.61 | **53.80** |
| | 1.5 | Uniform | 15.37 | 32.04 | 67.08 | 53.83 | 29.78 | 51.94 | 60.38 | **25.73** | **26.73** | 45.07 |
| | | PMQ | 11.05 | 18.31 | **71.65** | 53.58 | 30.63 | **57.19** | **64.64** | 24.12 | 26.56 | 46.91 |
| | | GEMQ | **9.15** | **16.64** | 71.55 | **58.50** | **32.08** | 56.40 | 62.90 | 25.39 | 24.05 | **47.27** |
| Qwen1.5-MoE (14.3B - A2.7B) | 16.0 | FP | 7.22 | 10.01 | 80.52 | 69.23 | 44.11 | 77.21 | 68.59 | 35.24 | 61.31 | 62.32 |
| | 2.5 | Uniform | 10.36 | 20.67 | 71.49 | 49.07 | 29.44 | 53.98 | 55.96 | 23.65 | 25.25 | 44.12 |
| | | PMQ | 9.01 | 13.57 | 77.58 | 60.56 | 36.26 | 70.69 | 65.82 | 28.68 | 42.38 | 54.57 |
| | | GEMQ | **8.02** | **12.25** | **78.40** | **66.96** | **42.92** | **71.78** | 65.82 | **30.62** | **52.14** | **58.38** |
| | 2.0 | Uniform | 14.25 | 34.59 | 66.38 | 42.89 | 30.29 | 49.06 | 53.12 | 21.78 | 23.92 | 41.06 |
| | | PMQ | 10.47 | 16.29 | 74.76 | 59.05 | 34.56 | 64.34 | 64.72 | **28.41** | 41.01 | 52.41 |
| | | GEMQ | **8.79** | **14.44** | **76.39** | **61.36** | **36.69** | **67.00** | **68.11** | 27.20 | **47.37** | **54.87** |
| | 1.5 | Uniform | 50.73 | 169.65 | 58.32 | 33.8 | 25.09 | 38.19 | 51.38 | 21.81 | 23.16 | 35.96 |
| | | PMQ | 14.10 | 23.88 | 70.46 | 55.77 | 31.91 | 54.55 | 62.51 | **24.99** | 32.88 | 47.58 |
| | | GEMQ | **12.69** | **21.63** | **74.81** | **58.71** | **35.41** | **58.65** | **63.38** | 23.38 | **39.50** | **50.55** |
| Qwen3-30B-A3B (30.5B - A3.3B) | 16.0 | FP | 8.71 | 14.06 | 80.25 | 79.29 | 55.80 | 77.75 | 71.11 | 58.89 | 77.88 | 71.57 |
| | 2.5 | Uniform | 10.29 | 17.45 | 78.07 | **75.93** | **50.07** | 71.72 | 68.01 | 42.52 | **72.72** | 65.58 |
| | | PMQ | 10.94 | 17.01 | 76.61 | 68.43 | 45.99 | 72.13 | 66.22 | 36.92 | 67.39 | 61.96 |
| | | GEMQ | **9.39** | **15.19** | **78.35** | 74.16 | 49.49 | **75.00** | **68.59** | **43.18** | 71.04 | **65.69** |
| | 2.0 | Uniform | 10.97 | 19.05 | 76.93 | **68.27** | **44.78** | 69.36 | **68.75** | **35.33** | **65.53** | **61.28** |
| | | PMQ | 13.91 | 19.84 | 73.18 | 53.96 | 35.84 | 70.05 | 64.17 | 24.22 | 38.29 | 51.39 |
| | | GEMQ | **10.34** | **16.77** | **76.93** | 65.11 | 43.77 | **72.22** | 68.11 | 34.34 | 63.21 | 60.53 |
| | 1.5 | Uniform | 21.09 | 47.80 | 69.21 | 57.74 | **37.88** | 53.44 | 62.98 | **25.43** | 39.25 | 49.42 |
| | | PMQ | 20.85 | 34.59 | 58.32 | 34.64 | 24.23 | **62.72** | **65.98** | 21.91 | 26.76 | 42.08 |
| | | GEMQ | **12.16** | **20.46** | **74.37** | **58.59** | 35.07 | 61.20 | 63.14 | 25.06 | **49.69** | **52.45** |
| Mixtral-8×7B (46.7B - A12.9B) | 16.0 | FP | 3.84 | 7.40 | 83.68 | 83.42 | 59.73 | 83.99 | 76.32 | 41.78 | 67.87 | 70.97 |
| | 2.5 | Uniform | 6.10 | 10.35 | 80.03 | 75.93 | **52.73** | 79.09 | 73.56 | 34.84 | **62.26** | **65.49** |
| | | PMQ | 5.10 | 9.21 | 80.36 | 74.87 | 51.28 | 79.18 | **73.95** | **34.94** | 55.78 | 64.34 |
| | | GEMQ | **5.03** | **9.02** | **81.07** | **75.93** | 51.79 | **79.57** | 73.72 | 34.10 | 59.74 | 65.13 |
| | 2.0 | Uniform | 6.29 | 11.73 | 75.14 | 64.73 | 43.77 | 71.31 | 67.72 | 29.15 | 38.23 | 55.72 |
| | | PMQ | 6.10 | 11.36 | **78.29** | **73.15** | **49.66** | **73.55** | **71.35** | **30.65** | 46.03 | **60.38** |
| | | GEMQ | **6.03** | **10.89** | 77.53 | 71.42 | 46.50 | 72.40 | 70.40 | 30.39 | **51.24** | 59.98 |
| | 1.5 | Uniform | 10.67 | 25.39 | 65.61 | 56.10 | 34.13 | 55.77 | 61.88 | 24.99 | 33.66 | 47.45 |
| | | PMQ | 8.47 | 20.77 | 73.01 | 63.13 | **37.37** | **64.16** | 65.98 | **27.04** | 31.76 | 51.78 |
| | | GEMQ | **7.93** | **16.20** | **73.67** | **63.47** | 36.95 | 59.93 | **67.25** | 26.20 | **36.50** | **52.00** |

**Implementation Details.** For bit-width allocation, we follow Huang et al. (2024a) and use the general-domain text dataset C4 for gradient extraction, sampling 128 random sequences of 2048 tokens each. For quantization, we follow prior work (Li et al., 2024; Huang et al., 2024a) to quantize all attention modules to 4 bits and assigning each expert a bit-width from $\{1, 2, 3\}$ according to our allocation results. In our experiments, we use *bits per expert* (bpe) to denote the expert quantization budget, defined as bpe = (total bits assigned) / (number of experts in the model). We consider three settings: 2.5, 2.0, and 1.5 bpe. Since our method employs post-quantization fine-tuning, router parameters are kept in 16-bit precision. For a fair comparison with existing methods, we also report results with routers quantized to 4 bits after fine-tuning in Appendix (Tab. 15). We employ group-wise asymmetric GPTQ quantization (group size 128), using 128 random sequences of length 2048 from the WikiText2 training set for calibration. For router fine-tuning, we use the same calibration set as in quantization. We use the AdamW optimizer (Loshchilov & Hutter, 2017) to minimize cross-entropy loss with learning rate $1e{-}4$, batch size 1, and weight decay $1e{-}4$, while keeping all other training settings identical to those used in pre-training. Fine-tuning is performed using PyTorch's `bfloat16` data type for one epoch. All experiments are conducted on three NVIDIA H100-80GB GPUs.

## 5.1 COMPARISON OF MoE-LLMs QUANTIZATION METHODS

We present a comprehensive comparison of mixed-precision MoE-LLMs quantization methods under various bit-budget settings. We compare against a uniform quantization baseline in which all experts

Table 2: Performance comparison of Mixtral-8×7B quantization using different calibration datasets. Perplexity↓ on the WikiText2 and C4 test sets, average accuracy↑ (%) across seven zero-shot tasks, and accuracy↑ (%) on GSM8K are reported.

| Method | WikiText2↓ | C4↓ | Zero-shot Avg.↑ | GSM8K↑ |
|---|---|---|---|---|
| FP | 3.84 | 7.40 | 70.97 | 57.77 |
| **2.5 bits** | | | | |
| PMQ (C4) | 5.10 | 9.21 | 64.34 | 33.06 |
| GEMQ (C4) | **5.03** | **9.02** | 65.13 | 31.77 |
| PMQ (MATH+C4) | 5.20 | 9.25 | 65.23 | 41.93 |
| GEMQ (MATH+C4) | 5.18 | 9.17 | **66.47** | **42.30** |
| **2.0 bits** | | | | |
| PMQ (C4) | 6.10 | 11.36 | 60.38 | 19.48 |
| GEMQ (C4) | **6.03** | 10.89 | 59.98 | 12.89 |
| PMQ (MATH+C4) | 6.16 | 11.25 | 60.20 | **23.84** |
| GEMQ (MATH+C4) | **6.03** | **10.77** | **61.07** | 23.12 |

are assigned the same bit-width. For the 2.5/1.5-bit settings, following prior work (Li et al., 2024; Chen et al., 2025), we quantize experts in the first half of the layers to 3/2 bits and those in the second half to 2/1 bits, and report the results as "Uniform" in Tab. 1. We also compare with the state-of-the-art mixed-precision method PMQ (Huang et al., 2024a), based on the official codebase. For a fair comparison, we vary only the bit-widths of experts while keeping all attention modules fixed at 4 bits, and apply GPTQ (Frantar et al., 2022) with identical quantization settings across all methods. As shown in Tab. 1, mixed-precision approaches generally outperform the uniform quantization baseline, particularly in low-bit settings, highlighting the importance of accounting for heterogeneous expert importance. Compared with the layer-wise local allocation method PMQ, the proposed GEMQ leverages global expert importance and mitigates router distortion, leading to better performance.

In addition to general text modeling and commonsense tasks, we also evaluate performance on the challenging math-reasoning benchmark GSM8K (Cobbe et al., 2021). As shown in Tab. 2, when calibrating on the generic corpus C4 alone, both methods exhibit a noticeable accuracy drop, indicating a domain bias between the calibration data and the test data. This effect is more pronounced for our GEMQ because it does not heavily rely on heuristic regularization (see Appendix A.2 for further discussion). Nevertheless, this issue can be effectively mitigated by using more balanced and representative calibration data. Specifically, we add another 128 sequences from the MATH dataset (Hendrycks et al., 2021) and combine them with the original 128 C4 sequences for calibration and fine-tuning. As shown in Tab. 2, GEMQ substantially benefits from this simple mixed-data calibration setup: performance on the challenging GSM8K math-reasoning benchmark improves significantly, with only marginal increase in general text-modeling perplexity. More importantly, under this setting, GEMQ outperforms the local-based PMQ method on text-modeling and commonsense tasks, while matching its performance on GSM8K. This highlights the stronger optimization capability brought by global bit allocation.

Additional evaluations on the few-shot capabilities, as well as further comparisons against other MoE-LLM quantization methods, are provided in Appendix A.2 and Appendix A.3, respectively.

## 5.2 ABLATION STUDY

**Global Expert Bit-width Allocation.** In the previous section, we demonstrated the advantage of global expert allocation compared with its local counterpart. Here, we provide further analysis by visualizing the layer-wise distribution of assigned bit-widths in Fig. 4a. As shown, our method effectively leverages variations in layer importance, allocating more bits to critical layers and thereby achieving better performance. Moreover, the results reveal that layer importance varies across models and bit budgets, underscoring the need for our automatic LP-based formulation to achieve flexible and effective bit allocation. In Fig. 4b, we compare the proposed method with several alternative global expert bit-width allocation strategies. A naive extension of PMQ that directly applies layer-wise coefficients in a global LP suffers significant performance degradation because expert importance

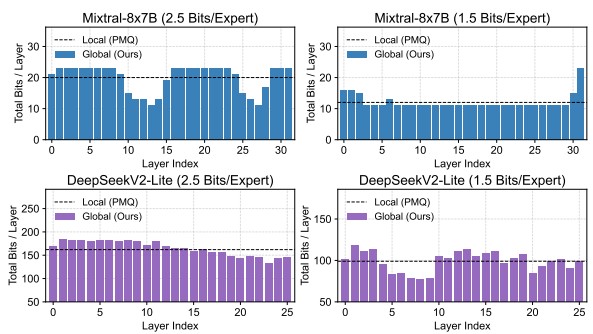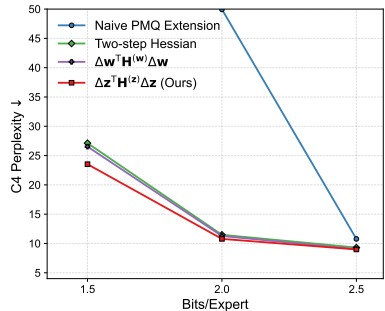

(a) Distribution of assigned bit-widths. Our method takes advantage of the layer-wise variation in expert importance.

(b) Comparison of global expert bit-width allocation methods on Mixtral-8×7B.

Figure 4: Analysis of global expert bit-width allocation.

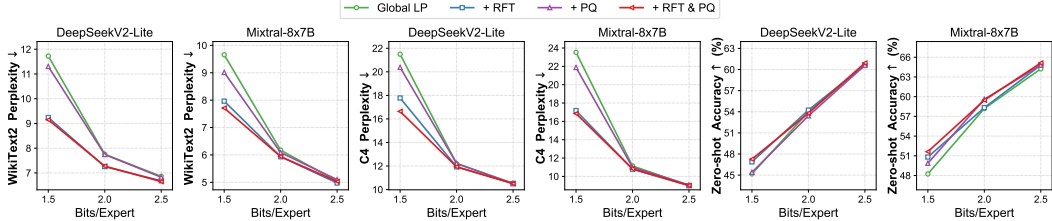

Figure 5: Ablation of the proposed techniques. "RFT" denotes global router fine-tuning, and "PQ" denotes progressive quantization. "Zero-shot Accuracy" is averaged over seven tasks.

calculated in this way is not comparable across layers. A two-step scheme that first assigns layer-level bit budgets and then allocates bits to experts yields better results but still underperforms global allocation methods. Finally, we evaluate a variant that does not use the approximation in Eq. 4. While this approach performs well in high-bit settings, it degrades significantly under low-bit quantization, suggesting that Eq. 4 offers a more reliable approximation for expert importance estimation.

Additionally, since our bit allocation formulation relies on gradients computed from a subset of calibration samples, variations in the sampled data may affect both the resulting allocation and the final performance. To examine this, we conduct robustness experiments using three random seeds to sample 128 sequences, as well as a larger subset of 2048 sequences, from both the C4 dataset and the mixed C4+MATH dataset for bit allocation. We compare the per-expert-per-bit estimated errors (*i.e.*, Eq. 6, the proxy for expert importance), the resulting expert-level bit assignments, and the layer-wise bit allocation statistics in Fig. 6, Fig. 9 and Fig. 10, along with the final performance in Tab. 5. The results show that GEMQ is relatively robust to sampling noise: expert importance is preserved across different calibration subsets, and model performance remains stable under different sampling choices.

Furthermore, we provide additional ablation experiments in Appendix A.5 to evaluate the use of extra constraints in our LP formulation and our choices of bit-width candidates.

**Global Router Fine-Tuning.** In Fig. 5, we compare our method with and without router fine-tuning after quantization. As shown, global router fine-tuning substantially reduces perplexity on both the WikiText2 and C4 test sets, particularly in the challenging 1.5 bits/expert setting where router logits are heavily distorted, highlighting the necessity of such calibration. Furthermore, we find that such calibration generalizes to downstream tasks, as evidenced by improvements in zero-shot accuracy. Tab. 3 compares our global router fine-tuning strategy with layer-wise calibration (Chen et al., 2025), which enforces routers to mimic the FP distribution. As seen, even when router logits in quantized models are replaced with recorded FP logits during inference, the performance gain is marginal, indicating that relying on the FP distribution does not yield optimal selection due to changes in expert weights. By contrast, fine-tuning routers yields substantial improvements, underscoring the effectiveness of task-loss-based global router adjustment after quantization. Additionally, we ablate the fine-tuning settings (*e.g.*, training epochs and number of calibration samples) and further analyze the effectiveness of fine-tuning, with several intriguing findings reported in Appendix A.6.

Table 3: Comparison of perplexity↓ for different router calibration methods on Mixtral-8×7B 1.5-bit quantization.

| Method | WikiText2 | C4 |
|---|---|---|
| FP | 3.84 | 7.40 |
| w/o Calibration | 9.66 | 23.53 |
| w/ FP Router Logits | 9.29 | 22.72 |
| w/ Router Fine-Tuned | **7.69** | **17.18** |

Table 4: Perplexity↓ and average router change ratio of 1.5-bit quantized Mixtral-8×7B, using different expert importance estimation models.

| Est. Model | WikiText2 | C4 | Change Ratio |
|---|---|---|---|
| FP | 9.66 | 23.53 | 41.31% |
| 4-bit | 9.39 | 22.92 | 39.10% |
| 2-bit | 9.33 | 22.80 | 34.69% |
| 2-bit (RFT) | **9.01** | **21.88** | 38.38% |

Table 5: Average performance of GEMQ-quantized models on three randomly sampled calibration subsets (128 sequences of 2048 tokens each) from the C4 dataset.

| #Bits | DeepSeekV2-Lite | | | Mixtral-8×7B | | |
|---|---|---|---|---|---|---|
| | WikiText2↓ | C4↓ | 0-shot Avg.↑ | WikiText2↓ | C4↓ | 0-shot Avg.↑ |
| 2.5 | $6.65 \pm 0.01$ | $10.47 \pm 0.02$ | $60.72 \pm 0.21$ | $4.95 \pm 0.02$ | $8.92 \pm 0.03$ | $65.02 \pm 0.13$ |
| 2.0 | $7.26 \pm 0.07$ | $11.80 \pm 0.04$ | $54.43 \pm 0.27$ | $5.89 \pm 0.03$ | $10.70 \pm 0.04$ | $60.00 \pm 0.21$ |
| 1.5 | $9.22 \pm 0.21$ | $17.09 \pm 0.26$ | $47.08 \pm 0.35$ | $7.75 \pm 0.15$ | $16.04 \pm 0.21$ | $52.79 \pm 0.33$ |

**Progressive Quantization.** We evaluate the effectiveness of the progressive quantization strategy in Fig. 5. As shown, this strategy improves both perplexity and zero-shot accuracy across models. Furthermore, combining progressive quantization with router fine-tuning yields additional performance gains on both models. As analyzed in Sec. 4.3, using a quantized model with a bit budget closer to the target may reduce expert selection changes (and thus abrupt changes in loss), since the two models lie closer in parameter space. We validate this in Tab. 4 and Fig. 11: when estimating expert importance for 1.5-bit quantization, using the 2-bit quantized model lowers the average expert selection change rate from 41.31% to 34.69%, compared to using the FP model. The reduction in abrupt changes in loss leads to more accurate expert importance estimation and improves the perplexity of the quantized model. Furthermore, fine-tuning the routers of the 2-bit quantized model and using it for expert importance estimation further improves performance, which is consistent with our analysis, as the model better satisfies the zero-gradient assumption and yields a more accurate approximation.

## 5.3 Memory Saving and Inference Efficiency

We follow PMQ (Huang et al., 2024a) and use the HQQ (Badri & Shaji, 2024) library to save quantized weights and perform dequantization. Detailed model size reduction and inference speedup comparisons across different quantized models are reported in Appendix A.4 (Tab. 11). Under the same bit budget, our GEMQ method reduces the model to the same size as PMQ, with only a negligible increase from storing router parameters in 16 bits. For inference speed, although routers in our method tend to select high-bit experts more frequently after router fine-tuning (see analysis in Appendix A.6), we observe only a minimal reduction in inference speed compared to PMQ.

## 6 Conclusion

In this work, we identify two key limitations in existing mixed-precision quantization methods for MoE-LLMs and propose GEMQ, which enables optimal expert bit-width allocation in pre-quantization and optimal router selection in post-quantization. We further introduce a progressive quantization framework that integrates both techniques to improve expert importance estimation in low-bit scenarios. GEMQ stands as a pioneering framework that validates the potential of global expert importance for mixed-precision MoE quantization. By formulating the allocation problem globally, GEMQ unlocks a substantially richer optimization space than local methods. Our empirical results provide compelling evidence that global allocation is a promising and effective approach for optimizing the compression-performance trade-off in MoE-LLMs.

REPRODUCIBILITY STATEMENT

To facilitate reproducibility, we detail the experimental setup, baseline methods, evaluation protocols, and training procedures in Sec. 5 and Appendix A.1, and present the final outputs of our method in Appendix A.8. The source code, scripts, and full configurations will be released to enable exact replication of our results. All datasets used in this study are publicly available and properly cited, ensuring that other researchers can readily access and validate our work.

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

# A  APPENDIX

## A.1  DETAILS ON MODELS AND EVALUATION

Table 6: Details of MoE-LLMs used in evaluation. "Expert Prop." and "Router Prop." denote the percentage of experts and routers in the total number of parameters ("#Params"), respectively. For the "#Experts" column, we follow the convention (#Routed Experts + #Shared Experts) × #Layers. Qwen3-30B-A3B and Mixtral-8×7B contain only routed experts and no shared experts. For DeepSeekV2-Lite, the first dense layer is omitted in the #Layers count. Top-K experts are selected from the routed experts.

| Model | #Params. | Expert Prop. | Router Prop. | #Experts | Top-K |
|---|---|---|---|---|---|
| DeepSeekV2-Lite | 15.7B | 94.51% | 0.021% | (64+2)×26 | 6 |
| Qwen1.5-MoE | 14.3B | 92.82% | 0.021% | (60+4)×24 | 4 |
| Qwen3-30B-A3B | 30.5B | 94.95% | 0.041% | 128×48 | 8 |
| Mixtral-8×7B | 46.7B | 97.00% | 0.002% | 8×32 | 2 |

To comprehensively evaluate the proposed GEMQ for MoE-LLMs quantization, we select four representative MoE-LLMs with varying sizes, architectures, and characteristics: DeepseekV2-Lite (Dai et al., 2024), Qwen1.5-MoE-A2.7B (Yang et al., 2025), Qwen3-30B-A3B (Yang et al., 2025), and Mixtral-8×7B (Jiang et al., 2024). The details of these models are summarized in Tab. 6. Note that, except for Qwen3-30B-A3B, all models are only pre-trained for language modeling without supervised fine-tuning (SFT). Qwen3-30B-A3B undergoes both pre-training and post-training.

In addition to evaluating perplexity on general language modeling benchmarks, we evaluate different quantization methods on seven zero-shot tasks: PIQA (Bisk et al., 2020), ARC-Easy (Clark et al., 2018), ARC-Challenge (Clark et al., 2018), HellaSwag (Zellers et al., 2019), WinoGrande (Sakaguchi et al., 2021), MathQA (Amini et al., 2019), and MMLU (Hendrycks et al., 2020). We also perform 5-shot evaluation on MMLU to assess the in-context learning ability of the models. Moreover, we evaluate our method on the challenging GSM8K (Cobbe et al., 2021) benchmark to assess the mathematical reasoning ability of quantized models. All benchmark results are obtained using LM-Evaluation-Harness (v0.4.8) (Gao et al., 2024). We report `acc_norm` when available; otherwise, `acc` is reported.

## A.2  QUANTIZATION RESULTS ON CHALLENGING BENCHMARKS

Table 7: Comparison of 5-shot MMLU accuracy↑ (%). #Bits indicates the average bits per expert.

| #Bits | Method | DeepSeekV2-Lite | Mixtral-8×7B |
|---|---|---|---|
| 16.0 | FP | 58.16 | 70.37 |
| 2.5 | Uniform | **54.09** | 50.51 |
| | PMQ | 49.98 | 60.70 |
| | GEMQ | 53.58 | **63.21** |
| 2.0 | Uniform | 27.67 | 51.17 |
| | PMQ | **38.60** | 46.79 |
| | GEMQ | 35.00 | **54.55** |
| 1.5 | Uniform | 28.97 | 34.31 |
| | PMQ | 29.12 | 34.53 |
| | GEMQ | **29.70** | **36.76** |

**Comparison of Few-Shot General Understanding Performance.**  Fig. 7 shows the performance comparison of models on the 5-shot MMLU benchmark. As seen, our GEMQ method outperforms the uniform baseline and the state-of-the-art mixed-precision method PMQ at most cases, demonstrating that our quantized models effectively preserve knowledge, reasoning ability, and few-shot in-context learning capability across diverse academic and professional subjects.

**Comparison of Reasoning and Mathematical Performance.** Tab. 2 shows the performance of quantized models on the challenging math benchmark GSM8K. With C4 calibration data alone, our method experiences a larger accuracy drop than PMQ, with the degradation becoming more pronounced at lower bit-widths. This illustrates the limited preservation of mathematical ability in the quantized models. We attribute this to bias introduced by task-specific calibration data, as also reported in prior work (Bandari et al., 2024; Huang et al., 2024a; Hu et al., 2025; Chen et al., 2025). This bias is exacerbated in our method because we do not heavily rely on heuristic regularization (*e.g.*, the $\alpha, \beta, \gamma$ hyperparameters on LP coefficients or the fixed per-layer bit budget used in PMQ (Huang et al., 2024a)), and our improved global optimization tends to find better minima for the calibration objective, which leads to good general perplexity but does not necessarily improve performance on specific tasks.

Nevertheless, we find that this issue can be effectively mitigated with a simple mixed-data strategy. Specifically, we randomly sample another 128 sequences from the MATH dataset (Hendrycks et al., 2021) and combine them with the original 128 C4 sequences (256 sequences in total) for calibration and fine-tuning. As shown below, GEMQ substantially benefits from this mixed calibration setup: performance on the challenging GSM8K math-reasoning benchmark improves significantly, with only minimal increase observed in general text-modeling perplexity. More importantly, under this mixed-data setting, GEMQ outperforms the local-based PMQ method on text-modeling and commonsense tasks, while matching its performance on GSM8K. This proves the stronger optimization capability brought by global bit allocation, as well as the improved precision of our approximated objective enabled by the progressive quantization framework.

While constructing an optimal calibration set that fully represents all downstream tasks remains an open question, the results in Tab. 2 indicate that our method can better benefit from a more balanced and diverse calibration dataset. In future work, we plan to explore self-sampling strategies as in MoEQuant (Hu et al., 2025) and joint-distribution modeling as in MoQa (Zheng et al., 2025) to construct a more representative calibration set.

### A.3 ADDITIONAL COMPARISONS WITH STATE-OF-THE-ART METHODS

In this section, we provide additional comparisons with recent representative MoE-LLM quantization methods, EAQuant (Fu et al., 2025) and MoEQuant (Hu et al., 2025), as well as with a representative mixed-precision quantization method for dense LLMs, SpQR (Dettmers et al., 2023).

**Comparison with EAQuant (Fu et al., 2025).** EAQuant primarily focuses on outlier suppression under uniform bit-width weight-activation quantization, whereas our method derives an optimal mixed-precision strategy for weight-only quantization. Although EAQuant also studies the router distribution-shift issue, it adopts a layer-wise rigid alignment scheme, which yields only marginal gains (*e.g.*, $< 0.1$ perplexity improvement reported in their paper). In contrast, our global router fine-tuning strategy produces substantial improvements for quantized models, as demonstrated in Fig. 5. A detailed comparison between the two router alignment strategies is also provided in Tab. 3 and Fig. 7. Additionally, EAQuant focuses on higher-bit quantization regimes ($\geq 3$ bpe), whereas our method targets more aggressive low-bit quantization ($\leq 2.5$ bpe) to address the substantial memory footprint of expert weights in MoE-LLMs. Tab. 8 presents a comparison between EAQuant and the proposed GEMQ for Mixtral-8×7B quantization. Since there is no shared quantization configuration between EAQuant and our method, we compare our 2.5-bpe model (W2.5A16) and 3.0-bpe model (W3A16) with the closest available setting in EAQuant, namely W3A4. Since there are slight differences between the FP16 baselines reported in the original paper and our own measurements, we additionally report the percentage increase after quantization relative to the FP16 baseline, computed as: $\Delta = (\text{Quantized} - \text{FP16})/\text{FP16}$. The results demonstrate that 3-bpe mixed-precision GEMQ consistently outperforms EAQuant, while the 2.5-bpe model is on par with it.

**Comparison with MoEQuant (Hu et al., 2025).** MoEQuant focuses on constructing optimal calibration data for uniform weight-only MoE quantization via a self-sampling strategy. It also extends GPTQ with affinity-guided weighting to prioritize high-affinity tokens and reduce quantization error. Nevertheless, MoEQuant does not explore mixed-precision bit allocation or address the router distortion issue introduced by expert quantization. Similar to EAQuant, MoEQuant also operates in higher-bit regimes ($\geq 3$ bpe). Tab. 9 presents a comparison between EAQuant and the proposed

Table 8: Comparison between EAQuant and GEMQ on Mixtral-8×7B. (*) denotes results from the original paper.

| Method | WikiText2↓ | C4↓ | PIQA↑ | ARC-E↑ | ARC-C↑ | BoolQ↑ | Winogrande↑ | Avg.↑ |
|---|---|---|---|---|---|---|---|---|
| FP16* | 3.84 | 6.98 | 83.41 | 83.29 | 55.80 | 84.56 | 75.85 | 76.58 |
| EAQuant-W3A4* | 5.27 | 8.23 | 79.05 | 78.45 | 50.68 | 78.69 | 69.30 | 71.23 |
| (Δ%) | (+37.24) | (+17.91) | (-5.23) | (-5.81) | (-9.18) | (-6.94) | (-8.64) | (-6.98) |
| FP16 | 3.84 | 7.40 | 83.68 | 83.42 | 59.73 | 85.05 | 76.32 | 77.64 |
| GEMQ-W3A16 | 4.37 | 8.06 | 82.54 | 80.68 | 57.34 | 85.02 | 74.90 | 76.10 |
| (Δ%) | **(+13.78)** | **(+8.97)** | **(-1.36)** | **(-3.28)** | **(-4.00)** | **(-0.04)** | **(-1.86)** | **(-1.99)** |
| GEMQ-W2.5A16 | 5.03 | 9.02 | 81.07 | 75.93 | 51.79 | 80.09 | 73.72 | 72.52 |
| (Δ%) | (+30.99) | (+21.89) | (-3.12) | (-8.98) | (-13.29) | (-5.83) | (-3.41) | (-6.59) |

Table 9: Comparison between MoEQuant and GEMQ on Mixtral-8×7B. (*) denotes results from the original paper.

| Method | WikiText2↓ | C4↓ | BoolQ↑ | MathQA↑ | MMLU↑ | GSM8K↑ | Avg.↑ |
|---|---|---|---|---|---|---|---|
| FP16* | 3.84 | 6.87 | 85.23 | 42.41 | 70.50 | 65.88 | 66.01 |
| MoEQuant-W3A16* | 4.90 | 8.24 | 82.81 | 38.82 | 64.10 | 43.21 | 57.24 |
| (Δ%) | (+27.60) | (+19.94) | (-2.84) | (-8.46) | (-9.08) | (-34.41) | (-13.29) |
| FP16 | 3.84 | 7.40 | 85.05 | 41.78 | 70.97 | 57.77 | 63.89 |
| GEMQ-W3A16 | 4.37 | 8.06 | 85.02 | 38.63 | 64.63 | 49.66 | 59.49 |
| (Δ%) | **(+13.78)** | **(+8.97)** | **(-0.04)** | **(-7.54)** | **(-8.93)** | **(-14.04)** | **(-6.90)** |
| GEMQ-W2.5A16 | 5.03 | 9.02 | 80.09 | 34.10 | 59.74 | 42.30 | 54.06 |
| (Δ%) | (+30.99) | (+21.89) | (-5.83) | (-18.38) | (-15.82) | (-26.78) | (-15.39) |

**GEMQ for Mixtral-8×7B quantization.** We compare our 2.5-bpe model (W2.5A16) and 3.0-bpe model (W3A16) with the W3A16 setting in MoEQuant. We also report the percentage increase after quantization relative to the FP16 baseline, computed as: $\Delta = (\text{Quantized} - \text{FP16})/\text{FP16}$. The results in the table show that 3-bpe mixed-precision GEMQ consistently outperforms MoEQuant across all benchmarks.

**Comparison with SpQR (Dettmers et al., 2023).** SpQR targets fine-grained, sub-tensor-level mixed precision for dense LLMs by preserving a small subset of outlier weights in full precision (16-bit) while quantizing the remaining weights to low precision. We compare GEMQ with SpQR on the MoE-LLM Mixtral-8×7B using the same group size (128) and the same calibration data (128 sequences from WikiText-2). All attention modules are quantized to 4-bit, and bits are allocated to expert modules according to the target bit budget. As shown in Tab. 10, our method slightly outperforms SpQR in the higher-bit regime. Under extreme low-bit settings, although SpQR retains a small set of 16-bit parameters in each expert, the vast majority of parameters are quantized to 1-bit, causing the model to collapse. In contrast, our expert-level mixed-precision method maintains sufficient precision for critical experts and, combined with router fine-tuning, prevents collapse and preserves performance.

Table 10: Comparison with SpQR and GEMQ on Mixtral-8×7B.

| #Bits | Method | WikiText2↓ | C4↓ | Zero-shot Avg.↑ |
|---|---|---|---|---|
| 2.5 | SpQR | 5.40 | 9.35 | 64.92 |
| | GEMQ | **5.03** | **9.02** | **65.13** |
| 1.5 | SpQR | >1000 | >1000 | 31.87 |
| | GEMQ | **7.93** | **16.20** | **52.00** |

A.4 MEMORY SAVING AND INFERENCE EFFICIENCY

Tab. 11 reports the model size reduction and inference speedup achieved by different quantization methods. We use the same HQQ library as PMQ for bit packing/unpacking and dequantization. Consequently, under the same bit budget, the only difference in model size is that we store the fine-tuned routers in 16 bits rather than 4 bits. As shown in Tab. 11, because router parameters account for only a small fraction of the model, our quantized models are only slightly larger than those of PMQ. On the other hand, differences in inference speed arise solely from the frequency of expert selection; for instance, selecting a 3-bit expert more frequently may increase data transfer time

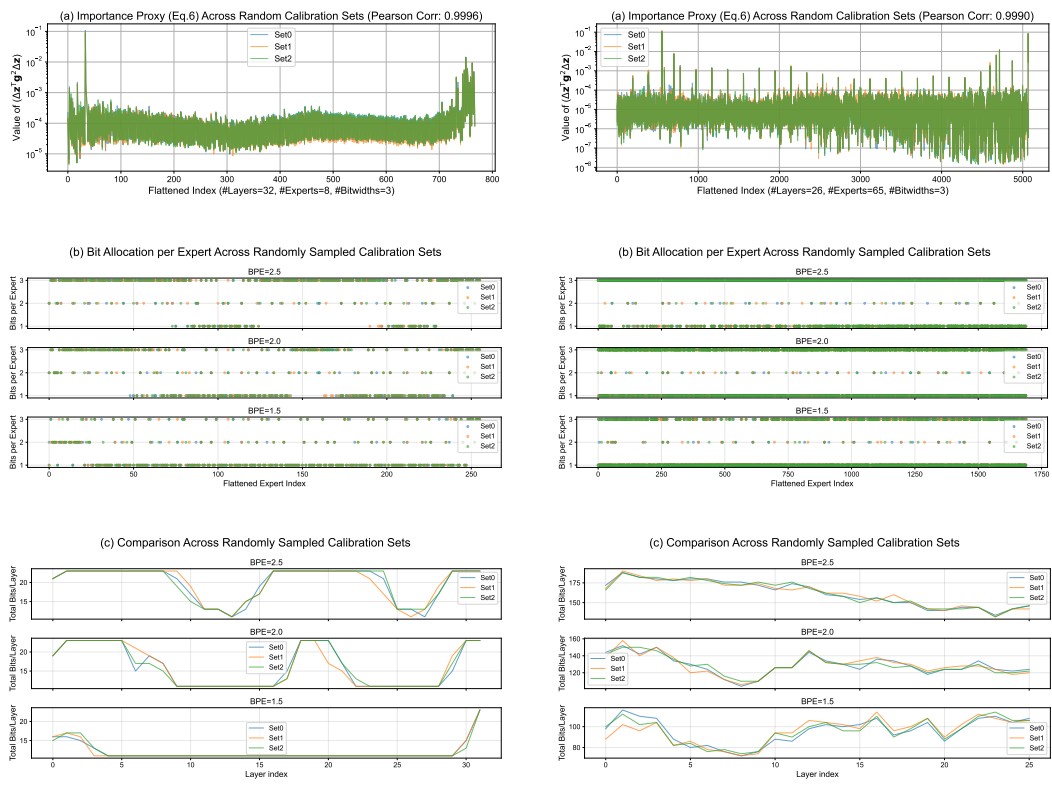

Statistics of Mixtral-8×7B.      Statistics of DeepSeekV2-Lite.

Figure 6: Statistics of Mixtral-8×7B (left) and DeepSeekV2-Lite (right) on three randomly sampled calibration subsets (128 sequences of 2048 tokens each) from the C4 dataset. Note that **dark green** in the figures indicates overlap.

and slow inference. On the WikiText2 test set, we observe only a minimal increase in inference time for our method compared with PMQ. We also discuss a potential direction for speed optimization in the next section.

Table 11: Memory reduction and inference speedup from quantization, with speed measured on the WikiText2 test set. Model sizes include quantization parameters (*i.e.*, scales and zero-points).

| #Bits | Method | Mixtral-8×7B | |
| --- | --- | --- | --- |
| | | Size (GB) | Speedup |
| 16.0 | FP | 86.99 | 1.00× |
| 2.5 | PMQ | 16.24 | 1.63× |
| | GEMQ | 16.31 | 1.60× |
| 1.5 | PMQ | 10.47 | 1.82× |
| | GEMQ | 10.51 | 1.77× |

## A.5 FURTHER ANALYSIS OF GLOBAL EXPERT BIT-WIDTH ALLOCATION

**Robustness Test for Bit-Width Allocation.** To examine how expert importance changes with different input subsets, we conduct robustness experiments using three random seeds to sample 128 sequences, as well as a larger subset of 2048 sequences, from both the C4 dataset and the mixed C4+MATH dataset for bit allocation. We compare the per-expert-per-bit estimated errors (*i.e.*, Eq. 6, the proxy for expert importance), the resulting expert-level bit assignments, and the layer-wise bit-allocation statistics in Fig. 6, Fig. 9 and Fig. 10. The correspding averaged performance is reported in Tab. 5. As shown the figures, GEMQ is relatively robust to sampling noise, as the estimated error

curves largely overlap even though only 128 sequences are used for calibration, achieving an average Pearson correlation over 0.99. Importantly, the key experts (*i.e.*, the peaks in the error-estimation curves) with large estimated errors are consistently identified across different samples. The overall expert-wise and layer-wise trends also remain closely aligned, indicating that expert importance is preserved across different calibration subsets. As a result, GEMQ yields consistent bit-allocation results and maintains model performance.

**Ablation on Extra Constraints in LP Formulation.**  We provide additional ablation experiments to evaluate the effect of adding different extra constraints to the LP formulation. Specifically, we compare the following settings on two bit budgets for Mixtral-8×7B bit allocation:

- *w/o Extra Constraints*: Using only the total bit-budget constraint, without any extra constraints;
- *Highest & 2nd Highest* (adopted in the paper): Each layer must be assigned at least one highest-bit and one second-highest-bit expert;
- *Only Highest*: Each layer must be assigned at least one highest-bit expert;
- *Only 2nd Highest*: Each layer must be assigned at least one second-highest-bit expert;
- *Highest Every 2 Layers*: Every two consecutive layers must be assigned at least one highest-bit expert.

As shown in Tab. 12, among all tested variants, enforcing at least one highest-bit and one second-highest-bit expert per layer yields the most favorable results. The benefit of adding this constraint is more pronounced in the 1.5-bit ultra-low-precision setting than in the 2.5-bit setting. We believe this stems from our analysis in Sec. 4.3 on the discontinuity of the MoE loss landscape, where the quantization error of an ultra-low-precision layer inevitably breaks the loss approximation. Incorporating the high-bit constraint keeps the optimization space within a mild-error region. As GEMQ contributes to resolving the loss approximation error, having such constraint in place also benefits our performance.

Table 12: Ablation of adding extra constraints in the global LP formulation on Mixtral-8×7B.

| Constraints | WikiText2↓ | C4↓ | Zero-shot Avg.↑ |
|---|---|---|---|
| **2.5 bpe** | | | |
| w/o Extra Constraints | 4.97 | 8.95 | 65.22 |
| Highest & 2nd highest | **4.95** | **8.92** | 65.17 |
| Only Highest | 4.98 | 8.94 | 65.09 |
| Only 2nd highest | 4.97 | 8.94 | **65.24** |
| Highest every 2 layers | 4.97 | 8.95 | 65.22 |
| **1.5 bpe** | | | |
| w/o Extra Constraints | 8.81 | 19.42 | 49.80 |
| Highest & 2nd highest | **7.92** | **16.13** | **53.42** |
| Only Highest | 8.02 | 16.87 | 51.31 |
| Only 2nd highest | 8.03 | 17.53 | 50.37 |
| Highest every 2 layers | 8.81 | 19.42 | 49.83 |

**Ablation on Expert Bit-width Candidates.**  In the main experiments, we use expert bit-width candidates $\{1, 2, 3\}$ and 4-bit attention to align with the prior mixed-precision work PMQ and ensure a fair comparison. Nevertheless, GEMQ is not restricted to this configuration, as the LP formulation naturally supports arbitrary and richer candidate sets. To demonstrate this flexibility, we provide experiments that expand the expert bit-width candidate set to $\mathcal{B} = \{0, 1, 2, 3, 4\}$ and evaluated additional attention-bit candidates $\{2, 4, 8\}$ on the Mixtral-8×7B model.

As shown in the Tab. 13, expanding the bit-candidate set leads to further improvement in our final ILP optimization objective and the perplexity on the C4 dataset, which is close to our calibration data distribution. These results indicate that our loss approximation and optimization algorithm are working as expected, where better minima can be found with a larger set of feasible solutions. Meanwhile, as a common tradeoff in LLM calibration, improved optimization toward the calibration

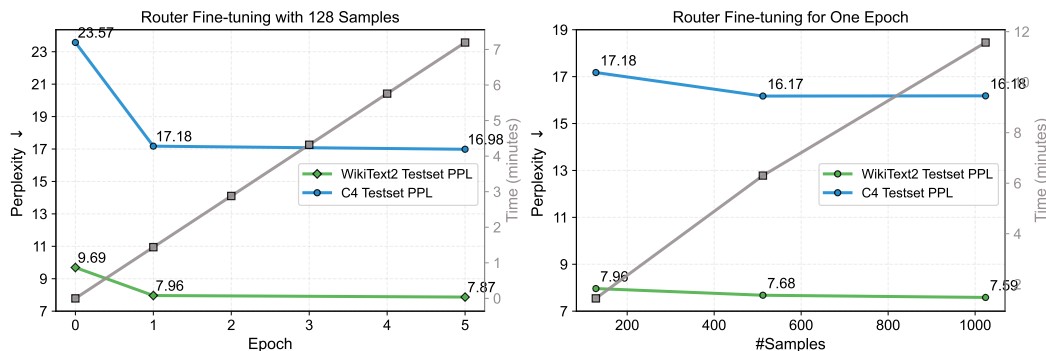

Figure 7: Ablation study on router fine-tuning settings. Weights are initialized from the quantized Mixtral-8×7B model (1.5 bits/expert). Training data are randomly extracted from the WikiText2 training set. Times are measured on three H100 GPUs.

set slightly hinders generalization performance on other tasks, which is more evident in the 1.5-bit regime. This issue can be resolved by employing a simple mixed-data strategy (as shown in Tab. 2) or exploring generalization objectives like sharpness-aware minimization in future work.

Table 13: Ablation of expert bit-width candidates on Mixtral-8×7B (attention bits = 4).

| Bits Per Expert | Bit Candidates $\mathcal{B}$ | Opt Obj (Eq.7)↓ | WT2↓ | C4↓ | Zero-shot Avg.↑ |
|---|---|---|---|---|---|
| 2.5 | {1,2,3} | 0.0144 | **4.97** | 8.95 | **65.22** |
| | {0,1,2,3} | 0.0139 | 5.02 | 8.91 | 64.96 |
| | {1,2,3,4} | 0.0138 | 5.00 | 8.95 | 65.19 |
| | {0,1,2,3,4} | **0.0131** | 5.06 | **8.90** | 65.12 |
| 1.5 | {1,2,3} | 0.0353 | **8.81** | 19.42 | **49.80** |
| | {0,1,2,3} | 0.0314 | 9.41 | 17.34 | 49.28 |
| | {1,2,3,4} | 0.0347 | 9.10 | 17.86 | 49.48 |
| | {0,1,2,3,4} | **0.0308** | 9.65 | **16.85** | 49.35 |

Table 14: Ablation of attention bit-width (bpe = 2.5, expert bit candidates $\mathcal{B}$ = {1,2,3}).

| Attention Bits | WikiText2↓ | C4↓ | Zero-shot Avg.↑ | Model Size (GB) |
|---|---|---|---|---|
| 8 | 4.81 | 8.63 | 66.44 | 17.00 |
| 4 | 4.97 | 8.95 | 65.22 | 16.37 |
| 2 | 31.14 | 82.79 | 35.16 | 16.06 |

**Ablation on Attention Bit-width.** We also evaluate different attention bit-widths, and the results are shown below. As observed, assigning lower bit-widths (*e.g.*, 2 bits) to the attention modules significantly degrades model performance, which is consistent with prior findings (Kim et al., 2023b; Li et al., 2024) that attention layers are more sensitive to quantization. In contrast, increasing the attention bit-width to 8 bits appears to be a favorable option, as it provides decent performance improvements with only a minimal increase in model size.

A.6 FURTHER ANALYSIS OF GLOBAL ROUTER FINE-TUNING

**Ablation Study on Router Fine-Tuning Settings.** We ablate the settings for global router fine-tuning in Fig. 7. As shown in the left figure, since routers contain only a small number of parameters, training converges within a single epoch in under 2 minutes. In the right figure, we observe that using more calibration samples can further reduce perplexity on the test set, but the improvement is marginal. We therefore use 128 samples in all experiments.

**Analysis of Routing Dynamics after Fine-Tuning.** As shown in Fig. 8, several intriguing patterns emerge after fine-tuning. In most cases, the assigned bit-widths of experts exhibit strong correlation with their activation frequencies after fine-tuning, indicating that router fine-tuning increases the

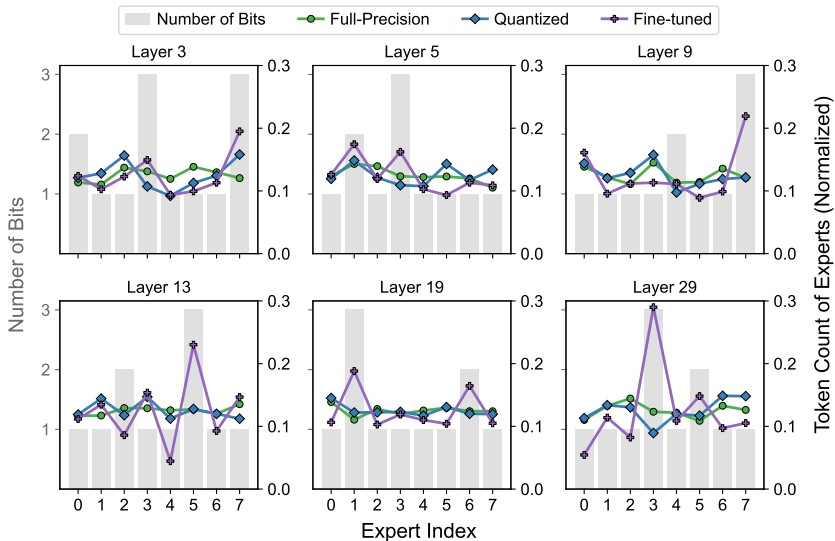

Figure 8: Comparison of router statistics from selected layers of the full-precision, quantized (1.5 bits/expert), and router fine-tuned Mixtral-8×7B models on the WikiText2 test set.

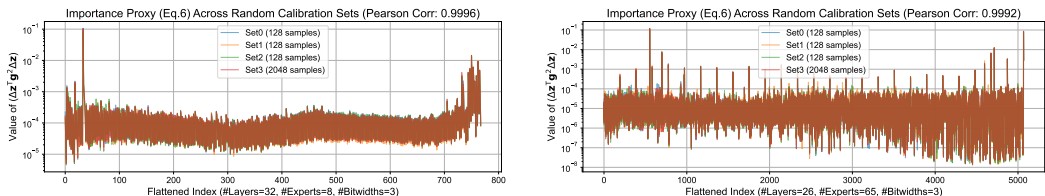

Figure 9: Statistics of Mixtral-8×7B (left) and DeepSeekV2-Lite (right) computed on four randomly sampled calibration subsets (each consisting of 2048-token sequences) from the C4 dataset. Note that **dark red** in the figures indicates overlap.

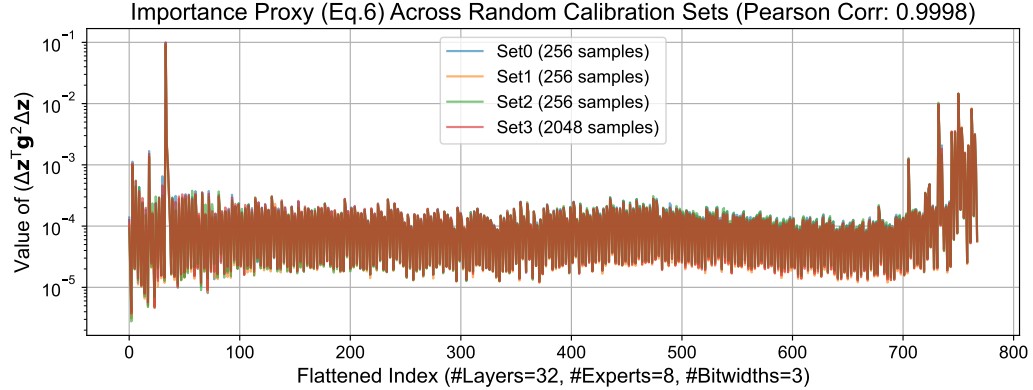

Figure 10: Statistics of Mixtral-8×7B computed on four randomly sampled calibration subsets (each consisting of 2048-token sequences) from the combined C4 and MATH dataset. Note that **dark red** in the figures indicates overlap.

selection of higher-bit experts (*e.g.*, layers 3, 5, and 19). This is intuitive as higher-bit experts incur less quantization loss and preserve more information. However, this is not always the case: low-bit experts can also contribute meaningfully to the model. For instance, in layer 9, although expert 4 has a higher bit width, the lower-bit expert 0 is activated more frequently. A potential issue arising

from this finding is the frequent use of high-bit experts, which disrupts load balance and increases inference time. However, we observe only a minimal decrease in inference speed compared to the baseline PMQ method. In practice, for deployment, high-bit experts can be treated as shared experts and handled separately for further optimization (Dai et al., 2024; Liu et al., 2024).

Another observation is that the router distribution after fine-tuning differs substantially from that of the FP model, implying that relying on the FP distribution does not lead to optimal selection after expert quantization. This suggests that the strategy of Chen et al. (2025), which enforces the router to mimic the FP distribution, may be suboptimal for expert-level mixed-precision quantization. Results in Tab. 8 further support this: when we replace the router logits of the quantized model with those from the FP model, the performance remains significantly worse than that of our global router fine-tuning method.

**Router Quantization after Fine-Tuning.** In our method, router weights are stored in 16-bit precision after fine-tuning. Since these weights constitute less than 0.04% of the total parameters, storing them in 16 bits introduces only a negligible increase in model size. However, prior work (Li et al., 2024; Huang et al., 2024a) typically quantizes router weights to 4 bits. For fair comparison, we also report results of quantizing the fine-tuned router weights to 4 bits after fine-tuning in Tab. 15. As shown, quantizing routers to 4 bits has little impact on model performance.

Table 15: Perplexity↓ of Mixtral-8×7B on WikiText2 and C4 w/ and w/o 4-bit router quantization after fine-tuning.

| #Bits | WikiText2 | | C4 | |
|---|---|---|---|---|
| | 16-bit Routers | 4-bit Routers | 16-bit Routers | 4-bit Routers |
| 2.5 | 5.03 | 5.04 | 9.01 | 9.02 |
| 2.0 | 5.94 | 5.94 | 10.82 | 10.82 |
| 1.5 | 7.71 | 7.72 | 16.85 | 16.82 |

## A.7 FURTHER ANALYSIS OF PROGRESSIVE QUANTIZATION

**Effectiveness of Progressive Quantization.** Tab. 16 and Tab. 17 provide ablation results comparing progressive quantization with computing model statistics from the full-precision model. As shown, progressive quantization consistently yields better performance, particularly in the low-bit regime. This finding aligns with our theoretical analysis in Sec. 4.3.

Table 16: Ablation of progressive quantization (PQ) on Mixtral-8×7B.

| #Bits | Method | WikiText2↓ | C4↓ | Zero-shot Avg.↑ |
|---|---|---|---|---|
| 2.5 | w/o PQ (from 16 bit) | 5.04 | 9.02 | 64.53 |
| | PQ (from 3 bit) | 5.03 | 9.01 | 64.82 |
| 2.0 | w/o PQ (from 16 bit) | 6.17 | 11.14 | 59.11 |
| | PQ (from 2.5 bit) | 6.09 | 11.00 | 59.60 |
| 1.5 | w/o PQ (from 16 bit) | 9.66 | 23.53 | 48.24 |
| | PQ (from 2.0 bit) | 9.01 | 21.88 | 49.87 |

Table 17: Ablation of progressive quantization (PQ) on DeepSeekV2-Lite.

| #Bits | Method | WikiText2↓ | C4↓ | Zero-shot Avg.↑ |
|---|---|---|---|---|
| 2.5 | w/o PQ (from 16 bit) | 6.86 | 10.54 | 60.48 |
| | PQ (from 3 bit) | 6.83 | 10.50 | 60.59 |
| 2.0 | w/o PQ (from 16 bit) | 7.75 | 12.20 | 54.00 |
| | PQ (from 2.5 bit) | 7.74 | 12.12 | 54.32 |
| 1.5 | w/o PQ (from 16 bit) | 11.72 | 21.49 | 45.17 |
| | PQ (from 2.0 bit) | 11.30 | 20.37 | 45.51 |

**Router Change Ratio Comparisons.** Fig. 11 shows the expert change ratio curves, from which the average ratios in Tab. 4 are computed. Each curve shows the router selection change ratio, computed by comparing the 1.5-bpe quantized model with the corresponding model used for expert importance estimation. As seen, using the FP model for 1.5-bit expert importance estimation is inaccurate, as large shifts in token-to-expert assignments induce abrupt loss changes. As we gradually use models with closer bit budgets for estimation, the expert selection change ratio decreases, indicating fewer abrupt loss changes and thus more accurate expert importance estimation. Intriguingly, when the 2-bit fine-tuned model is used for estimation, the resulting 1.5-bit model achieves better perplexity, even though the expert selection change ratio does not improve. We conjecture that this is due to the unstable and rugged loss landscape of the model after fine-tuning, which leads to a relatively high change ratio.

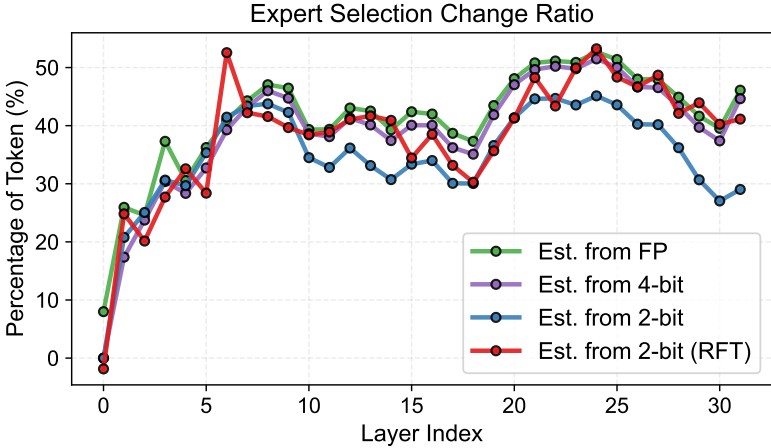

Figure 11: Router selection change ratio of Mixtral-8×7B computed on WikiText2 testset.

## A.8 Detailed Expert Bit-width Allocation Results

Detailed expert bit-width allocation results from our GEMQ method are shown in Fig. 12 to Fig. 15.

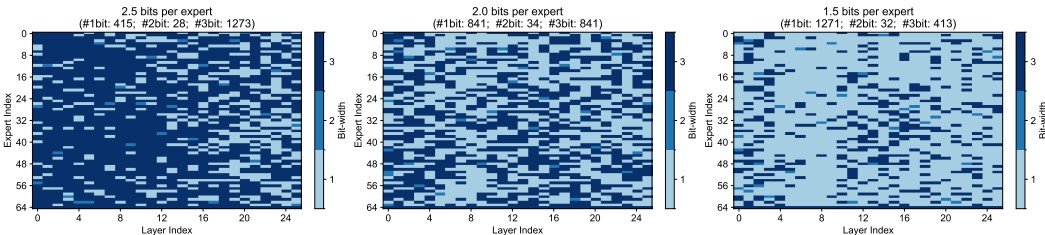

Figure 12: Bit-width allocation for DeepSeekV2-Lite; shared experts merged and shown last.

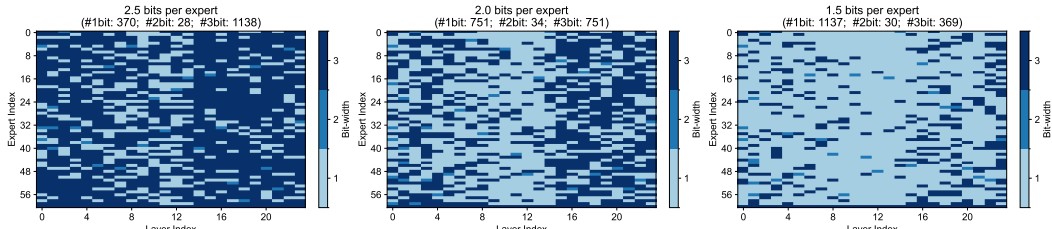

Figure 13: Bit-width allocation for Qwen1.5-MoE; shared experts merged and shown last.

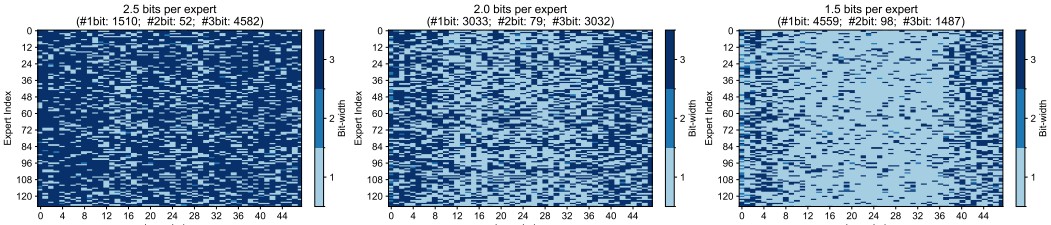

Figure 14: Bit-width allocation for Qwen3-30B-A3B.

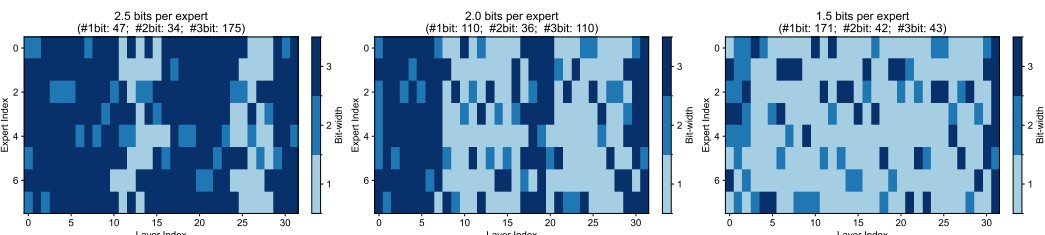

Figure 15: Bit-width allocation for Mixtral-8×7B.

## A.9 The Use of Large Language Models

We use large language models (ChatGPT-5) exclusively for paper writing refinement.

