# OpenReview forum: "Towards Global Expert-Level Mixed-Precision Quantization for Mixture-of-Experts LLMs"
_ICLR.cc/2026/Conference — Submitted to ICLR 2026_

### Official Review · Reviewer_Hrwz · 2025-10-15

**Soundness:** 2
**Presentation:** 3
**Contribution:** 2
**Rating:** 4
**Confidence:** 5

**Summary:**

This paper proposes GEMQ to tackle the high memory overhead of MoE-LLMs via global expert-level mixed-precision quantization. It addresses key flaws of existing methods (ignoring inter-layer importance differences and post-quantization router distortion) through three core components: global LP-based bit-width allocation, parameter-efficient router fine-tuning, and a progressive quantization framework.

**Strengths:**

1. The method proposed in the paper delivers practical acceleration gains, achieving a speedup of over 1.6× with negligible impact on accuracy.
2. The paper is well-written with a complete and coherent logical chain.

**Weaknesses:**

See "Questions" below.

**Questions:**

1. Figure 1 shows that loss gradients vary across layers, leading to the claim that layers differ in importance. However, the significance of this observation is highly dependent on input data. It remains unclear whether the inter-layer importance differences are consistent across different inputs, or if they vary with input changes. If inter-layer importance does differ across inputs, how should the bit-width of different layers be allocated?

2. The paper mentions using "task loss gradients" to globally measure expert importance, but it does not specify the **exact object of gradient calculation**—whether it refers to the gradients of expert weights or the gradients of layer outputs.

3. The paper sets a constraint that "each layer must contain at least one 3-bit expert and one 2-bit expert" to avoid information bottlenecks. However, it fails to explain why the combination of 3-bit/2-bit is chosen (instead of other combinations like 2-bit/1-bit) and lacks ablation experiments to verify the constraint’s impact on performance.

4. The paper only compares its method with PMQ as a baseline. However, there are many mixed-precision quantization methods for LLMs (e.g., SpQR). It is not addressed whether these methods also fail when applied to MoE models.

5. The ablation experiment in Table 2 shows that accuracy improves significantly after gate fine-tuning. Without fine-tuning, the model’s PPL is worse than that of PMQ. This indicates that the paper’s performance gains rely on fine-tuning, while the mixed-precision effect of the pre-quantization stage is weaker than that of PMQ.

6. The proposed method appears to be a combination of engineering tricks: extending intra-layer mixed-precision to inter-layer mixed-precision and adding fine-tuning after quantization. Both techniques are common in engineering practice.

7. Robustness tests are missing, and the reproducibility of results needs to be enhanced.


Based on the above issues, I would recommend a "Reject" decision. I would be happy to revise my rating if the authors can address these concerns in the revised manuscript.

---

> ### Author Response · Authors · 2025-11-21
> **Response to Reviewer (Part1)**
>
> We thank the reviewer for acknowledging the practical acceleration achieved by our method with minimal accuracy loss, as well as the clarity and coherence of the paper’s presentation. Below, we provide detailed point-by-point responses, with all relevant revisions incorporated into the manuscript.
>
>
> > Q1: Figure 1 shows that loss gradients vary across layers, leading to the claim that layers differ in importance. However, the significance of this observation is highly dependent on input data. It remains unclear whether the inter-layer importance differences are consistent across different inputs, or if they vary with input changes. If inter-layer importance does differ across inputs, how should the bit-width of different layers be allocated?
>
> Thank you for the insightful comment.
> To examine how expert importance changes with different input subsets, we conducted experiments using three different random seeds to sample 128 sequences from the C4 dataset for bit allocation. We compare the per-expert-per-bit estimated errors (i.e., Eq. 6, the proxy for expert importance), the resulting expert-level bit assignments, and the layer-wise bit-allocation statistics in *Figure 6* in Appendix A5, along with the final performance and its standard deviation in *Table 5* of the revised manuscript.
>
> As shown in the figures, GEMQ is relatively robust to sampling noise, as the estimated error curves largely overlap even though only 128 sequences are used for calibration, achieving an average Pearson correlation over 0.99. Importantly, the key experts (i.e., the peaks in the error-estimation curves) with large estimated errors are consistently identified across different samples. The overall expert-wise and layer-wise trends also remain closely aligned, indicating that **expert importance is preserved across different calibration subsets** and that **GEMQ produces consistent bit-allocation results**.
>
> We also provide a summary of the average performance with sample standard deviation in the table below. The results show that GEMQ stably maintains model performance across different calibration subsets.
>
> **Table: Average Performance of GEMQ on Random Calibration Subsets**
>
> |Model| BPE |  WT2$^\downarrow$ |C4$^\downarrow$ | 0-shot Avg$^\uparrow$ |
> |-|-------|-------------------------|-------------------------|-------------------------------|
> |DeepSeekV2-Lite| 2.5 | 6.65 ± 0.01 | 10.47 ± 0.02 | 60.72 ± 0.21 |
> || 2.0 | 7.26 ± 0.07 | 11.80 ± 0.04 | 54.43 ± 0.27 |
> || 1.5 | 9.22 ± 0.21 | 17.09 ± 0.26 | 47.08 ± 0.35 |
> |Mixtral-8×7B| 2.5 |4.95 ± 0.02 | 8.92 ± 0.03 | 65.02 ± 0.13 |
> ||2.0|5.89 ± 0.03 | 10.70 ± 0.04 | 60.00 ± 0.21 |
> ||1.5 |7.75 ± 0.15 | 16.04 ± 0.21 | 52.79 ± 0.33 |
>
>
> > Q2: The paper mentions using "task loss gradients" to globally measure expert importance, but it does not specify the exact object of gradient calculation—whether it refers to the gradients of expert weights or the gradients of layer outputs.
>
> We apologize for the confusion. In the paper, we use $\mathbf{z}$ to denote layer outputs and $\mathbf{g}^{(\mathbf{z})}$ to denote their gradients with respect to the task loss, as stated in *Line 226*: $\mathbf{g}_i^{(\mathbf{z})}$ represents the gradients of the corresponding layer outputs.

---

> ### Author Response · Authors · 2025-11-21
> **Response to Reviewer (Part2)**
>
> > Q3: The paper sets a constraint that "each layer must contain at least one 3-bit expert and one 2-bit expert" to avoid information bottlenecks. However, it fails to explain why the combination of 3-bit/2-bit is chosen (instead of other combinations like 2-bit/1-bit) and lacks ablation experiments to verify the constraint’s impact on performance.
>
> Thank you for pointing this out. We have conducted additional ablation experiments to evaluate the effect of adding different extra constraints to the LP formulation. Specifically, we compare the following settings on two bit budgets for Mixtral-8×7B bit allocation:
>
> - `w/o Extra Constraints`: Using only the total bit-budget constraint, without any extra constraints;
> - `Highest & 2nd Highest` (**adopted in the paper**): Each layer must be assigned at least one highest-bit and one second-highest-bit expert;
> - `Only Highest`: Each layer must be assigned at least one highest-bit expert;
> - `Only 2nd Highest`: Each layer must be assigned at least one second-highest-bit expert;
> - `Highest Every 2 Layers`: Every two consecutive layers must be assigned at least one highest-bit expert.
>
> As shown in the table below, among all tested variants, enforcing **at least one highest-bit and one second-highest-bit expert per layer** yields the most favorable results. The benefit of adding this constraint is more pronounced in the 1.5-bit ultra-low-precision setting than in the 2.5-bit setting.
> We believe this stems from our analysis in Section 4.3 on the discontinuity of the MoE loss landscape, where the quantization error of an ultra-low-precision layer inevitably breaks the loss approximation.
> Incorporating the high-bit constraint keeps the optimization space within a mild-error region. As GEMQ contributes to resolving the loss approximation error, having such constraint in place also benefits our performance.
>
>
> **Table: Ablation of Extra Constraints on GEMQ Bit Allocation for Mixtral-8×7B**
>
> |BPE| Constraint Setting | WT2$^\downarrow$ | C4$^\downarrow$ | 0-shot Avg$^\uparrow$ |
> |-|--------------------|--------|--------|------------------|
> |2.5| w/o Extra Constraints | 4.97 | 8.95 | 65.22 |
> || Highest & 2nd Highest | **4.95** | **8.92** | 65.17 |
> || Only Highest | 4.98 | 8.94 | 65.09 |
> || Only 2nd Highest | 4.97 | 8.94 | **65.24** |
> || Highest Every 2 Layers | 4.97 | 8.95 | 65.22 |
> |1.5| w/o Extra Constraints | 8.81 | 19.42 | 49.80 |
> || Highest & 2nd Highest | **7.92** | **16.13** | **53.42** |
> || Only Highest | 8.02 | 16.87 | 51.31 |
> || Only 2nd Highest | 8.03 | 17.53 | 50.37 |
> || Highest Every 2 Layers | 8.81 | 19.42 | 49.83 |
>
> > Q4: The paper only compares its method with PMQ as a baseline. However, there are many mixed-precision quantization methods for LLMs (e.g., SpQR). It is not addressed whether these methods also fail when applied to MoE models.
>
> Thank you for suggesting additional baselines. We argue that mixed-precision methods developed for dense LLMs are **not applicable to the extreme-low-bit quantization setting** required for MoE-LLMs, for the following reasons:
> - Most existing mixed-precision methods for dense LLMs, such as LLM.int8() [1] and SpQR [2], operate at average bit-widths above 4. Their primary objective is "lossless" quantization that preserves model performance as much as possible. However, even 4-bit quantization is insufficient to fit typical MoE-LLMs, such as Mixtral-8×7B, on common commercial GPUs like the NVIDIA 4090 with 24 GB of memory.
> In contrast, our goal is to reduce the model size as aggressively as possible while keeping performance degradation minimal, thereby enabling practical deployment on commercial GPUs.
> - Additionally, these methods cannot address the MoE-specific router distortion that becomes severe under extreme low-bit quantization, whereas our method can largely recover the resulting performance degradation.
> - Furthermore, from the inference perspective, SpQR employs fine-grained sub-tensor-level mixed precision, which necessitates tailored GEMM kernel implementations and specialized hardware support. In contrast, our expert-level approach is much simpler and only requires standard GEMM kernels with dequantization support, making it readily deployable on conventional platforms.
>
>
> [1] Dettmers, Tim, et al. “LLM. int8 (): 8-bit Matrix Multiplication for Transformers at Scale." arXiv preprint arXiv:2208.07339 (2022).
>
> [2] Dettmers, Tim, et al. “SpQR: A Sparse-Quantized Representation for Near-Lossless LLM Weight Compression." arXiv preprint arXiv:2306.03078 (2023).

---

> ### Author Response · Authors · 2025-11-21
> **Response to Reviewer (Part3)**
>
> > Q5: The ablation experiment in Table 2 shows that accuracy improves significantly after gate fine-tuning. Without fine-tuning, the model’s PPL is worse than that of PMQ. This indicates that the paper’s performance gains rely on fine-tuning, while the mixed-precision effect of the pre-quantization stage is weaker than that of PMQ.
>
> Thank you for the observation. This is in fact an important motivation for our analysis of loss-approximation error and for introducing router fine-tuning and the progressive quantization scheme in GEMQ. Theoretically, solving a well-defined ILP with a larger feasible set (i.e., global allocation) should yield a solution no worse than the layer-wise formulation. This is the case for the 2.5 bpe scenario where the error estimation is reliable: our global allocation **consistently outperforms PMQ across all tested models without router fine-tuning**, as shown in the table below. However, as analyzed in Section 4.2 and 4.3, pushing toward lower precision significantly increases the error in loss approximation, which has prevented researchers from effectively leveraging flexible global bit allocation to obtain better quantization schemes.
> GEMQ resolves this issue with the proposed router fine-tuning and progressive quantization techniques, reinstating the superiority of global allocation in the low-bit setting.
>
>
> **Table: Effect of Global LP on High-bit Scenarios (BPE=2.5)**
>
> |Model| Method | WT2$^\downarrow$ | C4$^\downarrow$ | 0-shot Avg$^\uparrow$ |
> |-|--------|--------|--------|------------------|
> |Mixtral-8×7B| PMQ | 5.10 | 9.21 | 64.34 |
> || GEMQ (w/o R-FT) | **5.09** | **9.05** | **64.82** |
> |DeepSeekV2-Lite| PMQ | 6.95 | 10.74 | 59.60 |
> || GEMQ (w/o R-FT) | **6.83** | **10.50** | **60.59** |
> |Qwen1.5-MoE| PMQ | 8.98 | 13.42 | 54.57 |
> || GEMQ (w/o R-FT) | **8.28** | **12.46** | **57.55** |
> |Qwen3-MoE| PMQ | 10.94 | 17.01 | 61.69 |
> || GEMQ (w/o R-FT) | **9.54** | **16.11** | **63.78** |
>
>
> > Q6: The proposed method appears to be a combination of engineering tricks: extending intra-layer mixed-precision to inter-layer mixed-precision and adding fine-tuning after quantization. Both techniques are common in engineering practice.
>
> Thank you for the comment. We respectfully clarify that the proposed GEMQ is **not** a collection of engineering tricks. GEMQ is the first method to successfully leverage **global expert importance** for mixed-precision bit allocation in MoE-LLMs, achieving the theoretically expected gains even at ultra-low precision. These improvements stem from our detailed analysis of quantization loss approximation, which in turn motivates our well-integrated progressive quantization algorithm. Such contribution is also recognized by reviewer `mxSW`, `rBrH`, and `wRrM`.
>
> More specifically, GEMQ makes three core technical contributions:
> -  Prior MoE-LLM quantization methods [1-5] all operate in a per-layer manner, whereas GEMQ formulates and solves a global LP that minimizes the approximated task loss across all experts in all layers simultaneously. To the best of our knowledge, our global LP is the first to explore **global expert importance** for mixed-precision bit allocation in MoE-LLMs.
> - Prior work [4-5] addresses router distortion by strictly aligning router outputs with those of the full-precision model. We show that such rigid alignment is suboptimal and instead introduce a **global router fine-tuning** approach that is theoretically motivated (Section 4.3) and highly effective in practice.
> - The above two techniques are tightly integrated into a coherent framework through a **progressive quantization** strategy, grounded in clear motivation and theoretical insights. This design effectively mitigates inaccuracies in expert-importance estimation under ultra–low-bit quantization, which is a crucial regime for compressing MoE-LLMs.
>
> [1] Huang, Wei, et al. “Mixture Compressor for Mixture-of-Experts LLMs Gains More." arXiv preprint arXiv:2410.06270 (2024).
>
> [2] Duanmu, Haojie, et al. “MxMoE: Mixed-precision Quantization for MoE with Accuracy and Performance Co-Design." arXiv preprint arXiv:2505.05799 (2025).
>
> [3] Hu, Xing, et al. “MoEQuant: Enhancing Quantization for Mixture-of-Experts Large Language Models via Expert-Balanced Sampling and Affinity Guidance." arXiv preprint arXiv:2505.03804 (2025).
>
> [4] Chen, Yuanteng, et al. “EAC-MoE: Expert-Selection Aware Compressor for Mixture-of-Experts Large Language Models." arXiv preprint arXiv:2508.01625 (2025).
>
> [5] Fu, Zhongqian, et al. “EAQuant: Enhancing Post-Training Quantization for MoE Models via Expert-Aware Optimization." arXiv preprint arXiv:2506.13329 (2025).

---

> ### Author Response · Authors · 2025-11-21
> **Response to Reviewer (Part4)**
>
> > Q7: Robustness tests are missing, and the reproducibility of results needs to be enhanced.
>
> Thank you for the thoughtful suggestion. We would like to highlight that our code **produces fully reproducible quantized models and evaluation results given a fixed calibration dataset**. For robustness evaluation, we have added experiments using multiple randomly sampled calibration sets and reported the average and standard deviation of the final performance. These results show that GEMQ remains stable across different calibration samples.
> To ensure reproducibility, we will publicly release all code and experimental settings.

---

> ### Comment · Reviewer_Hrwz · 2025-11-24
>
> Thank you very much for the authors' detailed replies and efforts. However, I still have many questions regarding the content of the paper. Building on the answers to each previous question, I would like to raise further doubts as follows:
>
> Q1: Feasibility of obtaining robust gradients using only 128 samples from C4
> Although the authors have provided certain explanations, I still harbor doubts about the ability to obtain stable gradients with merely 128 samples from C4. According to the analysis of MoE's inference behavior, the activation frequencies of different experts vary significantly. It is highly likely that some experts are not activated at all or only activated by an extremely small number of tokens with 128 samples. There remains a lack of clear evidence to demonstrate whether the gradients are sufficiently robust under such sparse activation scenarios.
> Furthermore, experts that are low-frequency activated on C4 may become high-frequency ones on other datasets. Relevant works such as Moqa have also pointed out the phenomenon that expert activation distributions are prone to mismatch across tasks. Therefore, I hope to further clarify: how to ensure the stability and generalization of gradient estimation with such a small calibration set?
>
> Q2: Why not directly use the gradients of expert weights to measure importance?
> The authors currently use output gradients to measure expert importance, but I still have questions. Output gradients are the comprehensive effect of multiple modules, not a direct indicator of the expert itself; in contrast, the gradients of the expert weights themselves are more direct metrics for measuring the expert’s own importance.
> In addition, after large-scale language models converge, the expectation of gradients is generally considered to be close to 0 (for example, GPTQ also approximates the first-order term to 0 based on this). Therefore, are the observed gradient differences in the experiments mainly due to the distribution bias of the calibration data rather than the true importance of the experts? I hope the authors can further explain this point.
>
> Q3: Regarding the constraint of forcing one 3-bit expert per layer
> I am still confused about the constraint of forcing one 3-bit expert per layer. If the proposed evaluation of expert importance is deemed reasonable, then bits should be allocated according to this scoring. Adding additional constraints will inevitably offset the optimal solution of the mathematical model.
>
> Q4:
> Although many mixed-precision quantization methods are mainly designed for dense models, this does not mean they will become ineffective in MoE models, right?
> Moreover, the authors have been emphasizing the importance of routers, but routers account for a very small proportion of the model. It is completely feasible to not quantize this part at all.
>
> Q6: The paper claims to be the first to use the global LP method for mixed-precision allocation of global expert importance. However, global LP does not seem directly reliable, or at least not a global optimal solution (otherwise, there would be no need to add the constraint that each layer must have some 3-bit experts to ensure performance).

---

> ### Author Response · Authors · 2025-11-26
> **Reply to Reviewer Hrwz [1/4]**
>
> We thank the reviewer for the continued engagement and the opportunity to clarify the technical details of our work. We are more than happy to provide further explanation and discussion.
>
> Before addressing specific questions, we would like to clarify **a fundamental misunderstanding regarding the proxy used for expert importance in Eq. 6**.
> While our method involves gradient computation, we **do not** use the raw gradient $\mathbb{E}[\mathbf{g}]$ itself as the indicator of importance. Instead, our goal is to measure the **sensitivity of the task loss to quantization noise** (i.e., the expected loss increase $\Delta \mathcal{L}$ caused by quantization). We achieve this via a second-order approximation where the importance is determined jointly by the perturbation errors $\Delta \mathbf{z}$ and the Hessian matrix (approximated via diagonal Fisher Information $\mathbb{E}[\mathrm{diag}(\mathbf{g}_z \mathbf{g}_z^\top)]$). This approximation has also been adopted and validated in prior quantization methods [1-2].
>
> > Reply1: Feasibility of obtaining robust gradients using only 128 samples from C4. How to ensure the stability and generalization of gradient estimation with such a small calibration set?
>
> We thank the reviewer for raising these critical questions. Below, we address these concerns from the perspectives of both stability and generalization.
>
> **Stability**. First, we demonstrate that 128 sequences (each containing 2048 tokens) are statistically sufficient to represent the distribution of the full dataset and yield a robust importance proxy.
>
> To more rigorously assess statistical stability, we increased the sampling size to 2048 sequences (4.2M tokens) from C4 when computing the importance proxy. As shown in Figure 9 of the revised manuscript, the proxy curve derived from 2048 sequences still closely aligns with that from 128 sequences for both models, yielding an average Pearson correlation above 0.99.
> Additionally, we repeat the experiment using a mixed dataset (128 sequences from C4 and 128 sequences from MATH [6]). As shown in Figure 10, we again observe a close alignment between the importance curves computed from 2048 sequences and those computed from 256 sequences.
>
> The large-scale verification above provides sufficient statistical significance and demonstrates the stability and robustness of our expert importance proxy, even when using only 128 randomly sampled sequences from the full dataset.
> We also note that using 128 sequences from C4 for computing expert statistics is a standard practice in prior MoE quantization and pruning literature [3-5].
>
> Regarding the sparse activation issue, we argue that our importance formulation handles this case well even when expert activations are imbalanced. For instance, if an expert is never activated, its contribution to the layer output $\Delta\mathbf{z}$ is zero, and it naturally receives a zero importance score. If an expert is rarely activated but produces significant changes in the layer output when it is used, it can still be assigned a high importance.
>
> **Generalization**. Then, we would like to address whether using C4 is sufficient to generalize to other downstream tasks.
>
> Our empirical results in Table 1 show that, using **the same 128 C4 calibration sequences**, our method outperforms both the uniform baseline and the local PMQ method on all **generic text modeling tasks** (perplexity on the WikiText2 and C4 test sets) and on most **general reasoning and linguistic tasks** (accuracy on zero-shot benchmarks). These results demonstrate that **using 128 C4 calibration sequences can generalize across a wide range of downstream tasks**.
>
> On the other hand, we agree with the reviewer that using C4 alone may not fully capture the distribution of **all downstream tasks**. We note that **such generalization challenge is inherent to all calibration-based quantization methods**.
> For domains with severe distribution shifts (e.g., mathematical reasoning), we show that simply mixing a small amount of domain-specific data (128 MATH sequences) with C4 is sufficient to recover performance (GSM8K accuracy increases from 31.77\% to 42.30\%, surpassing PMQ) while maintaining generic abilities (minimal degradation on text modeling and improvements on zero-shot benchmarks). Please see Table 2 for more details. While constructing an optimal calibration set that fully represents all downstream tasks remains an open question, the results above indicate that our method can **better benefit** from a more diverse calibration dataset.
> In future work, we plan to explore self-sampling strategies as in MoEQuant [7] and joint-distribution modeling as in MoQa [8] to construct a more representative calibration set.
>
> We have included a detailed discussion on the stability and generalization in Line 406-419, Line 464-470 and Appendix A2, and include a discussion on calibration dataset construction in Line 134-136 of the revised manuscript.

---

> ### Author Response · Authors · 2025-11-26
> **Reply to Reviewer Hrwz [2/4]**
>
> > Reply2: Why not directly use the gradients of expert weights to measure importance?
>
> Thank you for the detailed question. We would like to clarify both our choice of the importance proxy and the zero-gradient assumption.
>
> **Choice of importance metric**. As we clarify at the beginning of the response, the proposed proxy of expert importance is derived from sensitivity of the task loss to quantization noise, i.e., $\Delta\mathcal{L}$. **Gradients of the expert weights alone do not fully reflect $\Delta\mathcal{L}$ and cannot serve as a valid measure of expert importance**.
> Instead, our importance proxy for each expert consists of two components: (1) the perturbation term $\Delta\mathbf{z}$, which measures how quantizing a given expert affects the output of the entire layer; and (2) the approximated Hessian
> $\mathbf{H}_z \approx \mathrm{diag}(\mathbf{g}_z \mathbf{g}_z^\top)$ of the layer output, which reflects how changes in this layer propagate to and impact the final task loss $\Delta \mathcal{L}$. While another feasible approach is to compute the expert-wise approximation $\Delta \mathbf{w}^\top \mathbf{H}_w \Delta \mathbf{w}$ and approximate $\mathbf{H}_w$ using Fisher Information, we empirically find that using the layer output approximation $\Delta \mathbf{z}^\top \mathbf{H}_z \Delta \mathbf{z}$ yields slightly better results, as shown in Figure 4(b).
>
> **Clarification on zero gradients**. We fully agree with the reviewer that for a converged model, the expectation of the gradient is close to zero, i.e., $\mathbb{E}[\mathbf{g}] \approx \mathbf{0}$.
> In fact, we rely on this assumption to eliminate the first-order term in our derivation (Eq. 3).
> **The zero-gradient assumption also explains why gradients should not be used directly for measuring expert importance**.
> However, we respectfully clarify that zero gradient $\mathbb{E}[\mathbf{g}]$ (slope) **does not** imply zero Hessian (curvature) or zero Fisher Information $\mathbb{E}[\mathrm{diag}(\mathbf{g}\mathbf{g}^\top)]$. The second-order information we use remains meaningful under this assumption and continues to capture the local curvature of the loss landscape. It is worth noting that GPTQ is also based on second-order error estimation in its analysis.
>
> > Reply3: If the proposed evaluation of expert importance is deemed reasonable, then bits should be allocated according to this scoring.
>
> We apologize for the confusion and provide further clarification below.
>
> **Validity of the global formulation (high-bit regime)**. We respectfully clarify that adding constraints does not imply our Global LP formulation is unreliable. In fact, in the high-bit regime of 2.5 bpe, our plain global LP (without any additional constraints, router fine-tuning, or progressive quantization) consistently achieves better solutions than local methods PMQ, as shown in the table below. Additional evidence is provided in our earlier responses to Q3 and Q5. **These empirical results highlight the advantages of the global LP formulation, which imposes fewer restrictions and benefits from a larger solution space**.
>
> **Table: Effectiveness of plain global LP at 2.5 bpe (Mixtral-8×7B)**
> |Method|WT2$^\downarrow$|C4$^\downarrow$|0-shot Avg$^\uparrow$|
> |-|-|-|-|
> |Local PMQ| 5.10|9.21|64.34|
> |**Plain Global LP**|**5.03**|**8.98**|**65.08**|
>
> **Constraints as regularization (low-bit regime)**.
> On the other hand, we acknowledge that the plain importance proxy (Eq. 6) becomes less accurate in the ultra-low-bit regime (e.g., 1.5 bpe). This limitation arises from the second-order approximation of the loss change $\Delta \mathcal{L}$, which assumes small perturbations. As analyzed in Section 4.3, at extreme compression levels (e.g., 1-bit or 2-bit experts), the quantization noise becomes large enough to violate this assumption, resulting in less reliable importance estimates.
> In such cases, an unconstrained LP may produce degenerate solutions such as assigning an entire layer to 1-bit experts, which we empirically find to yield worse results.
> The constraint of "at least one high-bit expert" functions as a mild regularizer (in contrast to the strict "fixed total bits per layer" used in PMQ). It prevents the solver from entering regions where the approximation is known to be unstable, while still preserving a sufficiently large search space.
>
> In summary, we integrate the constraints, router fine-tuning, and progressive quantization into a holistic system, with each component contributing to mitigating the impact of less accurate importance estimates at low bit widths while still exploiting the larger and potentially more favorable solution space offered by the global LP.
>
> **Table: Effectiveness of full GEMQ at 1.5 bpe (Mixtral-8×7B)**
> |Method|WT2$^\downarrow$|C4$^\downarrow$|0-shot Avg$^\uparrow$|
> |-|-|-|-|
> | Local PMQ | 8.47 | 20.77 | 51.78 |
> | **Full GEMQ** | **7.93** | **16.20** | **52.00** |

---

> ### Author Response · Authors · 2025-11-26
> **Reply to Reviewer Hrwz [3/4]**
>
> > Reply4: Although many mixed-precision quantization methods are mainly designed for dense models, this does not mean they will become ineffective in MoE models, right?
> Moreover, the authors have been emphasizing the importance of routers, but routers account for a very small proportion of the model. It is completely feasible to not quantize this part at all.
>
> Thank you for the question.
> We fully agree with the reviewer that routers should not be quantized. However, we clarify that the **"Router Distortion" issue does not arise from quantizing the routers themselves, but rather from quantizing the experts**.
>
> For simplicity, let's consider one token $\mathbf{x}$ and one router at the $n$-th layer.
> Ideally, the router takes the input $\mathbf{x}$, computes the logits, and selects the most appropriate expert. However, two issues arise after quantizing the experts in the model:
>
> 1. **Input shift**: The token $\mathbf{x}$ arriving at layer $n$ is no longer the clean original input. It has accumulated quantization errors from the experts in the preceding $n-1$ layers, becoming a noisy version $\hat{\mathbf{x}}$.
> Since the FP16 router weights $\mathbf{W}_r$ are fixed, computing $\hat{\mathbf{x}}\mathbf{W}_r$ yields different logits than $\mathbf{x}\mathbf{W}_r$. This may cause the router to unintentionally select a different, suboptimal expert solely due to input noise. This is illustrated by the expert change ratios in Figure 1(b) and confirmed in our experiments.
>
> 1. **Expert capability shift**: Even if the input were perfect, the experts themselves in the $n$-th layer have changed after quantization. Suppose originally Expert A was the best and Expert B was second-best. In the quantized model, Expert A may be aggressively compressed to 1-bit (high error), while Expert B is preserved at 2-bit (low error), in which case Expert B may now be the better option. The original router doesn't know this and it will still send the token to the degraded Expert A.
>
> Consequently, if we simply apply existing quantization methods to quantize expert, the router distortion issue remains. Instead, our **router fine-tuning provides a crucial opportunity for the router to adapt and revise its logic to select the best experts under the new expert-quantized situation**.
>
> Following your suggestion, we compare GEMQ with the sub-tensor-level mixed-precision quantization method SpQR on Mixtral-8×7B, using the same group size (128) and the same calibration data (128 sequences from WikiText-2). We quantize all attention modules to 4-bit and allocate bits to expert modules according to the target bit budget.
> As shown in the table below, our method slightly outperforms SpQR in the higher-bit regime. Under extreme low-bit settings, although SpQR retains a small set of 16-bit parameters in each expert, the vast majority of parameters are quantized to 1-bit, causing the model to collapse. In contrast, our expert-level mixed-precision method maintains sufficient precision for critical experts and, combined with router fine-tuning, prevents collapse and preserves performance.
>
> **Table: Comparison with SpQR on Mixtral-8×7B**
> |BPE|Method|WT2$^\downarrow$|C4$^\downarrow$|0-shot Avg$^\uparrow$|
> |-|-|-|-|-|
> |2.5|SpQR|5.40|9.35|64.92|
> ||GEMQ|**5.03**|**9.02**|**65.13**|
> |1.5|SpQR|>1000|>1000|31.87|
> ||GEMQ|**7.93**|**16.20**|**52.00**|
>
> > Reply5: The paper claims to be the first to use the global LP method for mixed-precision allocation of global expert importance. However, global LP does not seem directly reliable, or at least not a global optimal solution.
>
> Thank you for the comment. We respectfully clarify that our use of the term "optimal" refers strictly to the mathematical property of the Integer Linear Programming (ILP) solver. Given the defined objective function (Eq. 7) and constraints, the ILP solver is guaranteed to find the global minimum.
> We acknowledge, however, that the expert importance proxy itself serves as **an approximation of the true task loss** and is not guaranteed to be optimal. Deriving a more accurate importance proxy at extremely low precision remains an open question and requires further investigation for future work.
>
> Again, we would like to highlight the core contribution of this work: **GEMQ stands as a pioneering framework that successfully validates the potential of global expert importance for mixed-precision MoE quantization**. By formulating the allocation problem globally, GEMQ unlocks a substantially richer optimization space than local methods.
> Our empirical results provide compelling evidence that global allocation is a promising and effective approach for optimizing the compression-performance trade-off in MoE-LLMs.
>
> We sincerely appreciate your time and effort for helping us improve this work. We hope the above responses address your concerns, and we look forward to any further clarification and discussion.

---

> > ### Author Response · Authors · 2025-11-26
> > **Reply to Reviewer Hrwz [4/4]**
> >
> > References
> >
> > [1] Li, Yuhang, et al. "BRECQ: Pushing the Limit of Post-Training Quantization by Block Reconstruction." arXiv preprint arXiv:2102.05426 (2021).
> >
> > [2] Kim, Sehoon, et al. "SqueezeLLM: Dense-and-Sparse Quantization." arXiv preprint arXiv:2306.07629 (2023).
> >
> > [3] Huang, Wei, et al. "Mixture Compressor for Mixture-of-Experts LLMs Gains More." arXiv preprint arXiv:2410.06270 (2024).
> >
> > [4] Duanmu, Haojie, et al. "MxMoE: Mixed-precision Quantization for MoE with Accuracy and Performance Co-Design." arXiv preprint arXiv:2505.05799 (2025).
> >
> > [5] Lu, Xudong, et al. "Not all experts are equal: Efficient expert pruning and skipping for mixture-of-experts large language models." arXiv preprint arXiv:2402.14800 (2024).
> >
> > [6] Hendrycks, Dan, et al. "Measuring Mathematical Problem Solving With the MATH Dataset." arXiv preprint arXiv:2103.03874 (2021).
> >
> > [7] Hu, Xing, et al. "MoEQuant: Enhancing Quantization for Mixture-of-Experts Large Language Models via Expert-Balanced Sampling and Affinity Guidance." arXiv preprint arXiv:2505.03804 (2025).
> >
> > [8] Zheng, Zihao, et al. "MoQa: Rethinking MoE Quantization with Multi-stage Data-model Distribution Awareness." arXiv preprint arXiv:2503.21135 (2025).

---

### Official Review · Reviewer_wRrM · 2025-10-16

**Soundness:** 2
**Presentation:** 2
**Contribution:** 2
**Rating:** 4
**Confidence:** 4

**Summary:**

To address the deployment memory bottleneck of large Mixture-of-Experts language models (MoE-LLMs), a Global Expert-level Mixed-Precision Quantization (GEMQ) method is proposed. Existing MoE-LLM quantization approaches face two key limitations: (1) they estimate expert importance locally within a single MoE layer, failing to capture global importance across layers, which leads to suboptimal bit allocation; (2) they ignore the impact of expert quantization on the router’s dynamic behavior, resulting in suboptimal routing. GEMQ addresses these challenges through three core designs: Global expert bit allocation: a linear programming model based on quantization error analysis to optimize bit distribution across all experts; Router-aware adaptation: efficient fine-tuning of router parameters accounting for less than 0.04% of total weights, ensuring compatibility with quantized experts; Progressive quantization framework: integrates the above two components and leverages fine-tuning from previous quantization rounds to improve expert importance estimation in low-bit scenarios. GEMQ achieves significant memory reduction and inference acceleration across various MoE-LLMs, while incurring minimal performance loss, outperforming existing methods such as PMQ.

**Strengths:**

1. The study clearly identifies two critical limitations of existing MoE quantization: local bit allocation and neglect of router dynamics, and experimentally validates their impact. Each GEMQ module is designed to directly address these shortcomings, forming a coherent logical framework.

2. Global bit allocation is formulated as a linear programming model based on task loss rather than heuristic coefficients, eliminating manual tuning and enabling adaptation to different MoE models. Router fine-tuning optimizes only a very small fraction of parameters.

3. The relationship between quantization error and task loss is derived using Taylor expansion and the Fisher information matrix (Eqs. 4–6), providing a theoretical foundation for global bit allocation. Additionally, GPTQ is adopted as the underlying quantization method, compatible with the HQQ library for model storage, and engineering considerations such as router quantization (Table 8) are incorporated, balancing theoretical rigor and practical feasibility.

**Weaknesses:**

1. Parameter Dependency and Progressive Framework Impact: The approach exhibits strong parameter dependency and is influenced by “previous-round model selection” in the progressive framework. If LP assigns too many low-bit experts to a layer, the router fine-tuning requires a higher learning rate to adapt; otherwise, router dynamics may drift more severely. However, the paper does not provide coordinated tuning rules across modules and relies solely on trial-and-error experiments (e.g., only one learning rate is tested). In practical applications, this would require traversing many parameter combinations, making tuning highly inefficient.

2. Limited Bit-Width Budget and Design Choices: The study only considers three bit budgets (2.5, 2.0, 1.5 bits/expert), which is a rather subjective design. It does not specify how iteration count (K value) or budget intervals are determined, nor does it analyze how different settings affect performance or efficiency. As a result, the method lacks clear guidance for parameter selection in different scenarios.

3. Performance on GSM8K and Calibration Sensitivity: On the GSM8K mathematical reasoning task, GEMQ performs significantly worse than the full-precision model, and its degradation is more severe under low-bit settings compared to PMQ. The paper attributes this to calibration-set fitting, suggesting that the quality of the MoE calibration set may be more critical than in mixed-precision setups. To ensure effective fine-tuning, GEMQ requires a strict match between calibration data and quantized data.

**Questions:**

1. Potential Optimization of Global Bit-Allocation Constraints: The linear programming model for global bit allocation imposes the constraint that “each layer must contain at least one 3-bit expert and one 2-bit expert” to avoid information bottlenecks. This constraint is adopted from the empirical setting in PMQ (Huang et al., 2024a) and has not been theoretically derived or validated via ablation studies. For small MoE models with very few layers or sparse experts (e.g., Qwen1.5-MoE with only 24 layers and 64 experts), this “fixed-type bit expert” constraint may waste bit budget. Would removing the constraint or changing it to “at least one high-bit expert every two layers” further reduce task loss under the same bit budget?

2. Undefined Failure Boundary of Taylor Approximation at Low Bits: The failure boundary of the Taylor expansion approximation under low-bit settings is not quantified. In Equation 4, the paper only mentions that “the approximation is unreliable at low bits” but does not define a specific threshold. For example, when the bit budget decreases from 2.0 bits to 1.5 bits, by how much does the Taylor approximation error increase? Is there a clear mathematical metric to determine in advance whether “the progressive framework should be enabled under the current bit budget”?

3. Theoretical or Empirical Justification for Calibration Synergy: Can theoretical or experimental evidence be provided to demonstrate synergy between the calibration set and this method—ensuring that experts are correctly selected, expert bit-widths are appropriately assigned, and post-calibration performance reaches optimal results across various tasks?

4. Can the method’s effectiveness be demonstrated on some multimodal MoE models, given that different modalities may have different impacts on expert selection?

---

> ### Author Response · Authors · 2025-11-21
> **Response to Reviewer (Part1)**
>
> We thank the reviewer for recognizing that our work identifies and addresses core limitations in existing MoE quantization. We also appreciate your acknowledgement of our principled and effective framework, along with its theoretical foundation. Below, we provide detailed point-by-point responses with all relevant revisions incorporated into the manuscript.
>
>
> > W1: Parameter Dependency and Progressive Framework Impact: The approach exhibits strong parameter dependency and is influenced by "previous-round model selection" in the progressive framework. If LP assigns too many low-bit experts to a layer, the router fine-tuning requires a higher learning rate to adapt; otherwise, router dynamics may drift more severely. However, the paper does not provide coordinated tuning rules across modules and relies solely on trial-and-error experiments (e.g., only one learning rate is tested). In practical applications, this would require traversing many parameter combinations, making tuning highly inefficient.
>
> Thank you for pointing out this practical concern. However, we respectfully disagree with the claim that our method "requires traversing many parameter combinations, making tuning highly inefficient."
> Our method only **fine-tunes all router weights simultaneously on a small calibration set**, which is a lightweight adjustment of the routing logits rather than a full-model optimization. In practice, the fine-tuning converges in about one epoch (see *Figure 7*) using **a single global learning-rate schedule shared across all routers**. Thus, it does not require per-layer tuning or extensive hyper-parameter searches. Across all experiments, **a fixed learning rate of 1e-4 consistently achieves stable and fast convergence**, demonstrating that the router fine-tuning stage is efficient and hyper-parameter robust.
>
>
> > W2: Limited Bit-Width Budget and Design Choices: The study only considers three bit budgets (2.5, 2.0, 1.5 bits/expert), which is a rather subjective design. It does not specify how iteration count (K value) or budget intervals are determined, nor does it analyze how different settings affect performance or efficiency. As a result, the method lacks clear guidance for parameter selection in different scenarios.
>
> Thank you for pointing this out. We would like to clarify that the selection of {2.5, 2.0, 1.5} follows the baseline method PMQ, which covers the bit-budget range [1.5, 2.5]. Yet our method is not limited to these choices and supports arbitrary bit budgets and interval settings. In practice, given a target bit budget, we can search for the quantized and fine-tuned model with the closest larger bit budget to compute the bit allocation.
>
> In principle, using a finer-grained interval for progressive quantization (i.e., reducing the interval size) can improve performance because the resulting models lie closer in parameter space, which leads to more accurate error estimation (see *Table 3* for an ablation on interval size). However, finer intervals also introduce additional computational overhead. Our empirical results show no significant gains when using intervals smaller than 0.5.

---

> ### Author Response · Authors · 2025-11-21
> **Response to Reviewer (Part2)**
>
> > W3: Performance on GSM8K and Calibration Sensitivity: On the GSM8K mathematical reasoning task, GEMQ performs significantly worse than the full-precision model, and its degradation is more severe under low-bit settings compared to PMQ. The paper attributes this to calibration-set fitting, suggesting that the quality of the MoE calibration set may be more critical than in mixed-precision setups. To ensure effective fine-tuning, GEMQ requires a strict match between calibration data and quantized data.
>
> Thank you for the insightful comment. Ideally, the calibration set should reflect the distribution of real-world data for all calibration-based quantization methods, including GEMQ. However, in our initial experiments, we relied solely on generic corpora for calibration and router fine-tuning, which does not adequately represent the target task distribution (e.g., math) and thus introduces domain bias. This bias is exacerbated in our method because we do not heavily rely on heuristic regularization (e.g., the $\alpha,\beta,\gamma$ hyper-parameters or the fixed per-layer bit budget used in PMQ), and our improved global optimization tends to find better minima for the calibration objective, which leads to good general perplexity but does not necessarily improve performance on specific tasks. This explains the observed degradation on math-reasoning tasks.
>
> Nevertheless, we find that this issue can be effectively mitigated with **a simple mixed-data strategy**. Specifically, we randomly sample another 128 sequences from the MATH dataset [1] and combine them with the original 128 C4 sequences (256 sequences in total) for calibration and fine-tuning.
> As shown below, GEMQ substantially benefits from this mixed calibration setup: performance on the challenging GSM8K math-reasoning benchmark improves significantly, with only minimal increase observed in general text-modeling perplexity.
> More importantly, under this mixed-data setting, **GEMQ outperforms the local-based PMQ method on text-modeling and commonsense tasks, while matching its performance on GSM8K**. This proves the stronger optimization capability brought by global bit allocation, as well as the improved precision of our approximated objective enabled by the progressive quantization framework.
>
> **Table: Ablation of Mixtral-8×7B Quantization with Different Calibration Datasets**
>
> |BPE| Method | WT2$^\downarrow$ | C4$^\downarrow$ | 0-shot Avg$^\uparrow$ | GSM8K$^\uparrow$ |
> |--|--------|--------|--------|------------------|------------|
> |2.5| PMQ (C4) | 5.10 | 9.21 | 64.34 | 33.06 |
> || GEMQ (C4) | **5.03** | **9.02** | 65.13 | 31.77 |
> || PMQ (MATH+C4) | 5.20 | 9.25 | 65.23 | 41.93 |
> || GEMQ (MATH+C4) | 5.18 | 9.17 | **66.47** | **42.30** |
> |2.0| PMQ (C4) | 6.10 | 11.36 | 60.38 | 19.48 |
> || GEMQ (C4) | **6.03** | 10.89 | 59.98 | 12.89 |
> || PMQ (MATH+C4) | 6.16 | 11.25 | 60.20 | **23.84** |
> || GEMQ (MATH+C4) | **6.03** | **10.77** | **61.07** | 23.12 |
>
> These results suggest that **both the calibration dataset and the mixed-precision strategy play important roles**, and that our method **benefits more** from a balanced calibration dataset than PMQ.
>
> [1] Hendrycks, Dan, et al. "Measuring Mathematical Problem Solving With the MATH Dataset." arXiv preprint arXiv:2103.03874 (2021).

---

> ### Author Response · Authors · 2025-11-21
> **Response to Reviewer (Part3)**
>
> > Q1: Potential Optimization of Global Bit-Allocation Constraints: The linear programming model for global bit allocation imposes the constraint that "each layer must contain at least one 3-bit expert and one 2-bit expert" to avoid information bottlenecks. This constraint is adopted from the empirical setting in PMQ (Huang et al., 2024a) and has not been theoretically derived or validated via ablation studies. For small MoE models with very few layers or sparse experts (e.g., Qwen1.5-MoE with only 24 layers and 64 experts), this "fixed-type bit expert" constraint may waste bit budget. Would removing the constraint or changing it to "at least one high-bit expert every two layers" further reduce task loss under the same bit budget?
>
> We appreciate this suggestion. We have conducted additional ablation experiments to evaluate the effect of adding different extra constraints to the LP formulation. Specifically, we compare the following settings on two bit budgets for Mixtral-8×7B bit allocation:
>
> - `w/o Extra Constraints`: Using only the total bit-budget constraint, without any extra constraints;
> - `Highest & 2nd Highest` (**adopted in the paper**): Each layer must be assigned at least one highest-bit and one second-highest-bit expert;
> - `Only Highest`: Each layer must be assigned at least one highest-bit expert;
> - `Only 2nd Highest`: Each layer must be assigned at least one second-highest-bit expert;
> - `Highest Every 2 Layers`: Every two consecutive layers must be assigned at least one highest-bit expert.
>
> As shown in the table below, among all tested variants, enforcing **at least one highest-bit and one second-highest-bit expert per layer** yields the most favorable results. The benefit of adding this constraint is more pronounced in the 1.5-bit ultra-low-precision setting than in the 2.5-bit setting.
> We believe this stems from our analysis in Section 4.3 on the discontinuity of the MoE loss landscape, where the quantization error of an ultra-low-precision layer inevitably breaks the loss approximation.
> Incorporating the high-bit constraint keeps the optimization space within a mild-error region. As GEMQ contributes to resolving the loss approximation error, having such constraint in place also benefits our performance.
>
> **Table: Ablation of Extra Constraints on GEMQ Bit Allocation for Mixtral-8×7B**
>
> |BPE| Constraint Setting | WT2$^\downarrow$ | C4$^\downarrow$ | 0-shot Avg$^\uparrow$ |
> |-|--------------------|--------|--------|------------------|
> |2.5| w/o Extra Constraints | 4.97 | 8.95 | 65.22 |
> || Highest & 2nd Highest | **4.95** | **8.92** | 65.17 |
> || Only Highest | 4.98 | 8.94 | 65.09 |
> || Only 2nd Highest | 4.97 | 8.94 | **65.24** |
> || Highest Every 2 Layers | 4.97 | 8.95 | 65.22 |
> |1.5| w/o Extra Constraints | 8.81 | 19.42 | 49.80 |
> || Highest & 2nd Highest | **7.92** | **16.13** | **53.42** |
> || Only Highest | 8.02 | 16.87 | 51.31 |
> || Only 2nd Highest | 8.03 | 17.53 | 50.37 |
> || Highest Every 2 Layers | 8.81 | 19.42 | 49.83 |

---

> ### Author Response · Authors · 2025-11-21
> **Response to Reviewer (Part4)**
>
> > Q2: Undefined Failure Boundary of Taylor Approximation at Low Bits: The failure boundary of the Taylor expansion approximation under low-bit settings is not quantified. In Equation 4, the paper only mentions that "the approximation is unreliable at low bits" but does not define a specific threshold. For example, when the bit budget decreases from 2.0 bits to 1.5 bits, by how much does the Taylor approximation error increase? Is there a clear mathematical metric to determine in advance whether "the progressive framework should be enabled under the current bit budget"?
>
> Thank you for the detailed question.
> We would like to clarify that the overall approximation error is not directly quantifiable, as different bit-allocation patterns can lead to different errors even under the same bit budget. That said, we cannot predetermine whether progressive quantization will be necessary at an arbitrary bit budget.
>
> The claim that "the approximation is unreliable at low bits" is based on our loss-landscape analysis for individual experts in MoE-LLMs (Section 4.3) and supported by the empirical results in the table below. These results show that **progressively computing model statistics consistently outperforms using full-precision statistics**, with the gap being most pronounced in the low-bit regime, confirming that FP16-based error estimates are unreliable. Overall, the findings suggest that progressive quantization is particularly beneficial when operating below 2 bpe.
>
>
> **Table: Ablation of Progressive Quantization (PQ) on Mixtral-8×7B**
> | BPE | Method | WT2$^\downarrow$ | C4$^\downarrow$ | 0-shot Avg$^\uparrow$ |
> |-----|---------|--------|--------|------------------|
> | 2.5 | w/o PQ (from FP16) | 5.04 | 9.02 | 64.53 |
> |     | **PQ (from 3.0-bpe)** | **5.03** | **9.01** | **64.82** |
> | 2.0 | w/o PQ (from FP16) | 6.17 | 11.14 | 59.11 |
> |     | **PQ (from 2.5-bpe)** | **6.09** | **11.00** | **59.60** |
> | 1.5 | w/o PQ (from FP16) | 9.66 | 23.53 | 48.24 |
> |     | **PQ (from 2.0-bpe)** | **9.01** | **21.88** | **49.87** |
>
>
> **Table: Ablation of Progressive Quantization (PQ) on DeepSeekV2-Lite**
> | BPE | Method | WT2$^\downarrow$ | C4$^\downarrow$ | 0-shot Avg$^\uparrow$ |
> |-----|---------|--------|--------|------------------|
> | 2.5 | w/o PQ (from FP16) | 6.86 | 10.54 | 60.48 |
> |     | **PQ (from 3-bpe)** | **6.83** | **10.50** | **60.59** |
> | 2.0 | w/o PQ (from FP16) | 7.75 | 12.20 | 54.00 |
> |     | **PQ (from 2.5-bpe)** | **7.74** | **12.12** | **54.32** |
> | 1.5 | w/o PQ (from FP16) | 11.72 | 21.49 | 45.17 |
> |     | **PQ (from 2.0-bpe)** | **11.30** | **20.37** | **45.51** |
>
>
> > Q3: Theoretical or Empirical Justification for Calibration Synergy: Can theoretical or experimental evidence be provided to demonstrate synergy between the calibration set and this method—ensuring that experts are correctly selected, expert bit-widths are appropriately assigned, and post-calibration performance reaches optimal results across various tasks?
>
> Thank you for the insightful question. We experimented with a more balanced calibration dataset that achieves a better trade-off than PMQ across general text modeling, commonsense QA, and mathematical reasoning tasks. Please see our response to your W3 for more details.
>
> > Q4: Can the method’s effectiveness be demonstrated on some multimodal MoE models, given that different modalities may have different impacts on expert selection?
>
> Thank you for the suggestion. We believe our method can be naturally extended to multi-modal MoE models for the following reasons. (1) In multi-modal models, inputs from different modalities are encoded into a shared embedding space before entering the MoE backbone. As a result, each MoE decoder layer receives a sequence of tokens without modality-specific distinctions during calibration or quantization, similar to standard LLMs. (2) Our method does not rely on any particular input modality. **The expert selection (bit allocation) decisions are entirely driven by the task loss**, and once the task loss is defined, the same mixed-precision allocation procedure can be applied.
>
> Although it is interesting to see how our method performs on existing multi-modal models, given the short rebuttal period and our limited experience with multi-modal models, we plan to attempt the experiments but cannot guarantee timely results.

---

> > ### Comment · Reviewer_wRrM · 2025-11-24
> >
> > Thank you for addressing most of my concerns. However, I still have the following questions:
> >
> > The authors concatenated the C4 and Math datasets to address the generalization issue in mathematical problem-solving. But for more general scenarios (e.g., reasoning or other linguistic tasks), how can the consistency between the quantized model and the floating-point model be ensured? If it is necessary to select and concatenate calibration datasets for re-quantization each time, how to resolve the costs incurred by repeated calibration?
> >
> > In the response to W1, the authors mentioned that no excessive fine-tuning cost is required. However, corresponding to Question 1 above, if a fixed learning rate of 1e-4 is used on a small calibration dataset, would adjustments be needed when expanding the calibration dataset? The claim that convergence can be achieved within one epoch with a fixed learning rate makes me question the necessity of fine-tuning the router—though this is merely an intuitive observation.
> >
> >
> >
> > Still regarding Question 1: Is it mandatory to retain experts with fixed bit-widths? Can reducing the proportion of low-precision experts decrease the share of high-bit-width experts? Personally, I suspect that such significant differences in bit-width configurations may lead to imbalances in calibration set errors during quantization or in deployment costs.

---

> ### Author Response · Authors · 2025-11-26
> **Reply to Reviewer wRrM [1/2]**
>
> Thank you for acknowledging our previous responses and for the practical follow-up questions. We are more than happy to provide further clarifications to your questions.
>
> > Reply1: The authors concatenated the C4 and Math datasets to address the generalization issue in mathematical problem-solving. But for more general scenarios (e.g., reasoning or other linguistic tasks), how can the consistency between the quantized model and the floating-point model be ensured? If it is necessary to select and concatenate calibration datasets for re-quantization each time, how to resolve the costs incurred by repeated calibration?
>
> Thank you for the thoughtful question. We would like to address the reviewer’s concerns regarding both the generalization capability of the calibration set and the potential cost of re-calibration under domain shifts.
>
> **Generalization in general scenarios**. First, we would like to emphasize that generalization challenges are inherent to **all calibration-based quantization methods**. While an ideal calibration set would fully cover the distribution of all downstream tasks, this is practically infeasible.
> Remarkably, however, we find that **strict domain alignment is not necessary** for our method. When calibrated solely on generic C4 sequences, the model already exhibits strong downstream performance. Moreover, when using a simple mixture of generic and math data, we observe gains **not only on math tasks but also on general reasoning and linguistic benchmarks**, as shown in the table below. Similar trends are observed across other bit-width settings and models. Thus, a single, diverse calibration set is sufficient for most common use cases.
>
> **Table: Mixtral-8×7B Performance at 2.0 bpe on Zero-Shot Reasoning and Linguistic Benchmarks**
> |Method|PIQA$^\uparrow$|ARC-Easy$^\uparrow$|ARC-Challenge$^\uparrow$|HellaSwag$^\uparrow$| WinoGrande$^\uparrow$|MathQA$^\uparrow$| MMLU$^\uparrow$|Avg.$^\uparrow$|
> |-|-:|-:|-:|-:|-:|-:|-:|-:|
> |PMQ (MATH+C4)|77.09|69.78|46.08|73.55|70.24| **35.18**|49.48|60.20|
> |GEMQ (C4)|77.53|71.42|46.50|72.40|70.40|30.39| **51.24**|59.98|
> |GEMQ (MATH+C4)|**79.33**|**72.56**|**47.53**|**74.63**|**70.64**|31.69|51.11| **61.07**|
>
> **Cost of re-calibration**.
> For rare cases where the existing calibration set may not fully represent the target domain, our method supports efficient **incremental re-calibration**. Since the calculation of expert importance (Eq. 6) is mathematically additive across data, we can simply compute statistics from a batch of new representative data, accumulate them into the existing statistics, and redo quantization. Importantly, this process is highly efficient. For instance, on Mixtral-8×7B, completing bit allocation, quantization, and fine-tuning for 128 new sequences takes less than one GPU hour.
>
> > Reply2: If a fixed learning rate of 1e-4 is used on a small calibration dataset, would adjustments be needed when expanding the calibration dataset? The claim that convergence can be achieved within one epoch with a fixed learning rate makes me question the necessity of fine-tuning the router—though this is merely an intuitive observation.
>
> We apologize for the confusion.
> We would like to clarify that router fine-tuning is a **lightweight yet critical** operation. To provide an intuitive explanation, we view experts as "knowledge modules" (the underlying information processors) and the router as a "dispatcher" that assigns tokens to appropriate experts.
>
> **Why it is lightweight**: During router fine-tuning, the **experts are completely frozen**. The goal is therefore not to learn new knowledge (which would be complex and slow), but simply to **re-allocate** tokens among the fixed experts. Since the task is limited to adjusting dispatch probabilities (changing <0.04% of the total model parameters), the optimization landscape is simple, leading to rapid convergence. We observe convergence within one epoch on both the WikiText2 (128 sequences) and the mixed MATH+C4 (256 sequences) calibration datasets using a learning rate of 1e-4, showing that it's sufficient for routers to see the calibration data only once.
>
> **Why it is critical**: Despite its simplicity, this step is indispensable. **Without fine-tuning, the "dispatcher" operates based on the original expert traits, ignoring the fact that these experts have been perturbed by quantization noise**. Such perturbations distort both the router inputs and each expert’s own behavior, shifting the model away from its original converged local minimum and degrading performance. Router fine-tuning provides a crucial opportunity for the router to adapt to these changes and revise its dispatch logic so that it selects the most appropriate experts under the quantized model.
> Ablation studies in Figure 5 and Table 3 confirm that removing router fine-tuning results in a significant performance drop, especially under low-bit cases where experts are quantized significantly.

---

> > ### Author Response · Authors · 2025-11-26
> > **Reply to Reviewer wRrM [2/2]**
> >
> > > Reply3: Is it mandatory to retain experts with fixed bit-widths? Can reducing the proportion of low-precision experts decrease the share of high-bit-width experts? Personally, I suspect that such significant differences in bit-width configurations may lead to imbalances in calibration set errors during quantization or in deployment costs.
> >
> > We interpret the reviewer’s concern as an inquiry into whether a more uniform bit allocation strategy (i.e., reducing the proportion of low-precision experts to lower the share of high-bit ones) would offer better performance. Please let us know if we have misunderstood the question.
> >
> > In fact, the "significant difference" in bit-widths is a deliberate design driven by the highly skewed sensitivity of MoE experts (see Figure 6). A uniform strategy (e.g., assigning all experts to 2-bit) is suboptimal because it degrades the few critical experts essential for performance. By contrast, our method preserves these key experts at 3-bit by aggressively compressing the non-critical majority to 1-bit.
> >
> > The effectiveness of such allocation is evident in Table 1, where **mixed-precision methods generally outperform uniform counterparts across low to high bit budgets**. Furthermore, regarding concerns about "calibration errors", the table below shows that uniformly assigning all experts to 2-bit results in a higher (worse) LP objective value compared to our mixed-precision strategy.
> >
> > **Table: ILP Optimization Objective Comparison**
> > | Method   | Opt Obj (Eq.7)$^\downarrow$ |
> > |--------------|--------------------|
> > | Uniform      | 0.0520             |
> > | Global LP    | 0.0236             |
> >
> > We sincerely appreciate your time and effort for helping us improve this work. We hope the above responses address your concerns, and we look forward to any further clarification and discussion.

---

### Official Review · Reviewer_rBrH · 2025-10-27

**Soundness:** 3
**Presentation:** 3
**Contribution:** 3
**Rating:** 6
**Confidence:** 5

**Summary:**

This paper proposes Global Expert-level Mixed-precision Quantization (GEMQ) for MoE-LLMs. The method estimates expert importance with a task-loss–motivated proxy and solves a global binary LP to allocate expert bit-widths under a budget. It then fine-tunes only router parameters to adapt routing after quantization, and further adopts a progressive quantization schedule that re-estimates importance from a nearby quantized-and-finetuned model. Experiments on DeepSeekV2-Lite, Qwen1.5-MoE, Qwen3-30B-A3B, and Mixtral‑8×7B show improved perplexity and competitive zero-shot accuracy under aggressive bit budgets, with large memory savings.

**Strengths:**

- The paper frames expert-bit assignment as a global optimization over all experts, rather than per-layer, yielding a principled resource–accuracy trade-off under a single LP.

- The empirical validation spans multiple MoE-LLMs (DeepSeekV2‑Lite, Qwen1.5‑MoE, Qwen3‑30B‑A3B, Mixtral‑8×7B) with consistent perplexity gains at 1.5–2.5 bpe, and strong memory reduction.

- The paper carefully analyzes inter-layer variations in expert importance and shows why a global allocator outperforms naive globalizations of local schemes.

- The paper is well written and easy to reproduce.

**Weaknesses:**

- Bit choices and the trade-off surface are narrow. Experiments constrain experts to {1, 2, 3} bits and attention to 4 bits which limits the observed efficiency–accuracy Pareto. There is little exploration of richer candidate sets (e.g., {0(prune), 1, 2, 3, 4}), mixed attention precisions, or hardware-aware constraints, making it hard to fully characterize cost–quality trade-offs.

- Importance is estimated on generic corpora (C4/WikiText2) and routers are fine-tuned with the same small calibration set. The paper itself notes worse GSM8K math results and attributes this to calibration bias toward general language modeling. This underscores a potential fragility: globally optimized bit layouts (and router readjustments) may overfit the calibration/task distribution and not transfer to domains with different expert usage.

- For Mixtral‑8×7B at 2.5 bpe, GEMQ’s average zero-shot accuracy (65.13) slightly trails the uniform schedule (65.49) despite improved perplexity (Table 1), suggesting that global LP + router FT does not uniformly dominate in accuracy at milder compression.

**Questions:**

- How does GEMQ behave with richer candidate sets, e.g., allowing 0-bit (expert skipping/pruning) or 4-bit experts, and varying attention precision?

- How robust are the LP coefficients to calibration sampling noise? Please report the variance of Eq. 6 costs across multiple random calibration draws and its impact on the final allocation and accuracy.

---

> ### Author Response · Authors · 2025-11-21
> **Response to Reviewer (Part1)**
>
> We thank the reviewer for recognizing our global optimization framework, the strength of our empirical validation, and the paper’s clarity and reproducibility. Below, we provide detailed point-by-point responses with all relevant revisions incorporated into the manuscript.
>
>
> > W1: Bit choices and the trade-off surface are narrow. Experiments constrain experts to {1, 2, 3} bits and attention to 4 bits which limits the observed efficiency–accuracy Pareto. There is little exploration of richer candidate sets (e.g., {0(prune), 1, 2, 3, 4}), mixed attention precisions, or hardware-aware constraints, making it hard to fully characterize cost–quality trade-offs.
>
> We appreciate the suggestion.
> We chose the expert bit-width candidates {1, 2, 3} and 4-bit attention to align with the prior mixed-precision work PMQ to ensure a fair comparison. Nevertheless, GEMQ is not restricted to this configuration, as the LP formulation naturally supports arbitrary and richer candidate sets. To demonstrate this flexibility, we have added new experiments that expand the expert bit-width candidate set to $\mathcal{B}$={0, 1, 2, 3, 4} and evaluated additional attention-bit candidates {2, 4, 8} on the Mixtral-8×7B model.
>
> As shown in the table below, expanding the bit-candidate set leads to further improvement in our final ILP optimization objective and the perplexity on the C4 dataset, which is close to our calibration data distribution. These results indicate that our loss approximation and optimization algorithm are working as expected, where better minima can be found with a larger set of feasible solutions. Meanwhile, as a common tradeoff in LLM calibration, improved optimization toward the calibration set slightly hinders generalization performance on other tasks, which is more evident in the 1.5-bit regime. This issue can be resolved by employing a simple mixed-data strategy (see our response to W2) or exploring generalization objectives like sharpness-aware minimization in future work.
>
>
> **Table: Ablation of Expert Bit-width Candidates on Mixtral-8×7B (attention bits=4)**
>
> | BPE | Bit Candidates $\mathcal{B}$ | Opt Obj (Eq.7)$^\downarrow$ | WT2$^\downarrow$ | C4$^\downarrow$ | 0-shot Avg$^\uparrow$ |
> |-----|----------------|------------------|--------|--------|------------------|
> | 2.5 | {1,2,3}         | 0.0144 | **4.97** | 8.95 | **65.22** |
> | | {0,1,2,3}       | 0.0139 | 5.02 | 8.91 | 64.96 |
> | | {1,2,3,4}       | 0.0138 | 5.00 | 8.95 | 65.19 |
> | | {0,1,2,3,4}     | **0.0131** | 5.06 | **8.90** | 65.12 |
> | 1.5 | {1,2,3}         | 0.0353 | **8.81** | 19.42 | **49.80** |
> | | {0,1,2,3}       | 0.0314 | 9.41 | 17.34 | 49.28 |
> | | {1,2,3,4}       | 0.0347 | 9.10 | 17.86 | 49.48 |
> | | {0,1,2,3,4}     | **0.0308** | 9.65 | **16.85** | 49.35 |
>
>
> We also evaluated different attention bit-widths, and the results are shown below. As observed, assigning lower bit-widths (e.g., 2 bits) to the attention modules significantly degrades model performance, which is consistent with prior findings [1-2] that attention layers are more sensitive to quantization. In contrast, increasing the attention bit-width to 8 bits appears to be a favorable option, as it provides decent performance improvements with only a minimal increase in model size.
>
>
> **Table: Ablation of Attention Bit-width (expert bits $\mathcal{B}$={1,2,3})**
>
> | BPE | Attention Bits | WT2$^\downarrow$ | C4$^\downarrow$ | 0-shot Avg$^\uparrow$ | Model Size (GB) |
> |-----|----------------|--------|--------|------------------|------------------|
> | 2.5 | 8 | 4.81 | 8.63 | 66.44 | 17.00 |
> | | 4 | 4.97 | 8.95 | 65.22 | 16.37 |
> | | 2 | 31.14 | 82.79 | 35.16 | 16.06 |
>
>
> [1] Kim, Young Jin, Raffy Fahim, and Hany Hassan Awadalla. "Mixture of Quantized Experts (MoQE): Complementary Effect of Low-bit Quantization and Robustness." arXiv preprint arXiv:2310.02410 (2023).
>
> [2] Li, Pingzhi, et al. "QuantMoE-Bench: Examining Post-Training Quantization for Mixture-of-Experts." arXiv preprint arXiv:2406.08155 (2024).

---

> ### Author Response · Authors · 2025-11-21
> **Response to Reviewer (Part2)**
>
> > W2: Importance is estimated on generic corpora (C4/WikiText2) and routers are fine-tuned with the same small calibration set. The paper itself notes worse GSM8K math results and attributes this to calibration bias toward general language modeling. This underscores a potential fragility: globally optimized bit layouts (and router readjustments) may overfit the calibration/task distribution and not transfer to domains with different expert usage.
>
> Thank you for the thoughtful comment. Ideally, the calibration set should reflect the distribution of real-world data for all calibration-based quantization methods, including GEMQ. However, in our initial experiments, we relied solely on generic corpora for calibration and router fine-tuning, which does not adequately represent the target task distribution (e.g., math) and thus introduces domain bias. This bias is exacerbated in our method because we do not heavily rely on heuristic regularization (e.g., the $\alpha,\beta,\gamma$ hyper-parameters or the fixed per-layer bit budget used in PMQ), and our improved global optimization tends to find better minima for the calibration objective, which leads to good general perplexity but does not necessarily improve performance on specific tasks. This explains the observed degradation on math-reasoning tasks.
>
> Nevertheless, we find that this issue can be effectively mitigated with **a simple mixed-data strategy**. Specifically, we randomly sample another 128 sequences from the MATH dataset [1] and combine them with the original 128 C4 sequences (256 sequences in total) for calibration and fine-tuning.
> As shown below, GEMQ substantially benefits from this mixed calibration setup: performance on the challenging GSM8K math-reasoning benchmark improves significantly, with only minimal increase observed in general text-modeling perplexity.
> More importantly, under this mixed-data setting, **GEMQ outperforms the local-based PMQ method on text-modeling and commonsense tasks, while matching its performance on GSM8K**. This proves the stronger optimization capability brought by global bit allocation, as well as the improved precision of our approximated objective enabled by the progressive quantization framework.
>
> **Table: Ablation of Mixtral-8×7B Quantization with Different Calibration Datasets**
>
> |BPE| Method | WT2$^\downarrow$ | C4$^\downarrow$ | 0-shot Avg$^\uparrow$ | GSM8K$^\uparrow$ |
> |--|--------|--------|--------|------------------|------------|
> |2.5| PMQ (C4) | 5.10 | 9.21 | 64.34 | 33.06 |
> || GEMQ (C4) | **5.03** | **9.02** | 65.13 | 31.77 |
> || PMQ (MATH+C4) | 5.20 | 9.25 | 65.23 | 41.93 |
> || GEMQ (MATH+C4) | 5.18 | 9.17 | **66.47** | **42.30** |
> |2.0| PMQ (C4) | 6.10 | 11.36 | 60.38 | 19.48 |
> || GEMQ (C4) | **6.03** | 10.89 | 59.98 | 12.89 |
> || PMQ (MATH+C4) | 6.16 | 11.25 | 60.20 | **23.84** |
> || GEMQ (MATH+C4) | **6.03** | **10.77** | **61.07** | 23.12 |
>
> While the construction of an optimal calibration set remains an open question and requires further exploration, the results above indicate that our method can **better benefit** from a more balanced calibration dataset.
>
> [1] Hendrycks, Dan, et al. "Measuring Mathematical Problem Solving With the MATH Dataset." arXiv preprint arXiv:2103.03874 (2021).
>
>
>
> > W3: For Mixtral-8×7B at 2.5 bpe, GEMQ’s average zero-shot accuracy (65.13) slightly trails the uniform schedule (65.49) despite improved perplexity (Table 1), suggesting that global LP + router FT does not uniformly dominate in accuracy at milder compression.
>
> Thank you for the observation. We find that **the advantages of mixed precision are most pronounced in the ultra-low-bit regime (e.g., ≤ 2 bpe)**. As the bit budget increases, the performance gap between mixed-precision and uniform quantization generally narrows, a trend also observed in prior mixed-precision methods such as PMQ.
> Nevertheless, **our method outperforms the uniform baselines in most cases under the 2.5-bpe setting**. The only exception is Mixtral-8×7B, which has only eight experts per layer (256 experts in total, whereas other models typically have more than 1000). We conjecture that this limited number of experts restricts the search space for mixed-precision bit assignment and consequently reduces the potential gains.
>
>
> > Q1: How does GEMQ behave with richer candidate sets, e.g., allowing 0-bit (expert skipping/pruning) or 4-bit experts, and varying attention precision?
>
> Please see our response to your W1.

---

> ### Author Response · Authors · 2025-11-21
> **Response to Reviewer (Part3)**
>
> > Q2: How robust are the LP coefficients to calibration sampling noise? Please report the variance of Eq. 6 costs across multiple random calibration draws and its impact on the final allocation and accuracy.
>
> Thank you for the valuable suggestion. We have added robustness experiments using three different random seeds for sampling 128 sequences from the C4 dataset for bit allocation. We report the per-expert-per-bit estimated errors (i.e., Eq. 6), the resulting expert-level bit assignments, and the layer-wise bit-allocation statistics in *Figure 6* in Appendix A5, along with the final performance and its standard deviation in *Table 5* of the revised manuscript. A copy of the table is shown below.
>
>
> **Table: Average Performance of GEMQ on Random Calibration Subsets**
>
> |Model| BPE |  WT2$^\downarrow$ |C4$^\downarrow$ | 0-shot Avg$^\uparrow$ |
> |-|-------|-------------------------|-------------------------|-------------------------------|
> |DeepSeekV2-Lite| 2.5 | 6.65 ± 0.01 | 10.47 ± 0.02 | 60.72 ± 0.21 |
> || 2.0 | 7.26 ± 0.07 | 11.80 ± 0.04 | 54.43 ± 0.27 |
> || 1.5 | 9.22 ± 0.21 | 17.09 ± 0.26 | 47.08 ± 0.35 |
> |Mixtral-8×7B| 2.5 |4.95 ± 0.02 | 8.92 ± 0.03 | 65.02 ± 0.13 |
> ||2.0|5.89 ± 0.03 | 10.70 ± 0.04 | 60.00 ± 0.21 |
> ||1.5 |7.75 ± 0.15 | 16.04 ± 0.21 | 52.79 ± 0.33 |
>
> As shown in the figures, GEMQ is relatively robust to sampling noise, as the estimated error curves largely overlap even though only 128 sequences are used for calibration, achieving an average Pearson correlation over 0.99. Importantly, the key experts (i.e., the peaks in the error-estimation curves) with large estimated errors are consistently identified across different samples. The overall expert-wise and layer-wise trends also remain closely aligned, indicating that expert importance is preserved across different calibration subsets. As a result, GEMQ yields consistent bit-allocation results and maintains model performance.

---

> ### Author Response · Authors · 2025-11-28
> **A Kind Reminder Regarding Our Responses to Your Comments**
>
> Dear Reviewer rBrH,
>
> Thank you once again for your time and valuable comments. As the discussion phase is approaching its end, we would like to kindly check whether our responses have adequately addressed your concerns. If there are any remaining questions or points that would benefit from further clarification, please feel free to let us know.
>
> We sincerely look forward to your feedback.
>
> Best regards,
>
> Authors of Paper 7529

---

### Official Review · Reviewer_mxSw · 2025-10-31

**Soundness:** 3
**Presentation:** 3
**Contribution:** 2
**Rating:** 4
**Confidence:** 3

**Summary:**

This paper proposes a Global Expert-level Mixed-Precision Quantization (GEMQ) framework for large-scale Mixture-of-Experts Language Models (MoE-LLMs).

It mainly consists of three core modules: Global Linear Programming (Global LP) bit allocation: jointly optimizing the bit width of all experts; Router Fine-tuning (RFT): lightweightly adjusting routing parameters after quantization to correct routing offsets; and Progressive Quantization: gradually reducing the bit width and quantizing in stages to ensure stability at low bit widths. Experiments on multiple models show results that outperform PMQ and uniform quantization methods.

**Strengths:**

1. This paper proposes a practical solution to address the memory and bandwidth bottlenecks faced by MoE-LLMs during deployment.
2. The logical relationships between the three modules (Global LP, RFT, and Progressive Quantization) are clearly defined, and the theoretical motivation and loss surface analysis are presented in Figure 3.
3. The paper covers multiple mainstream MoE architectures, and evaluations include perplexity, zero/few-shot accuracy, and various ablation experiments.
4. Routing fine-tuning (RFT) requires only less than 0.05% parameter adjustments to significantly restore performance at low bit precision.

**Weaknesses:**

1. Insufficient theoretical support: Although the method is based on linear programming (LP), it does not provide any form of theoretical analysis or upper bound on the error.

2. Lack of quantitative ablation in incremental quantization: The authors visualize the smoothness of the loss surface in the appendix, but do not provide quantitative indicators to explain the contribution or error accumulation at each stage.

3. Insufficient comparison with state-of-the-art methods: Only PMQ and Uniform are compared, without comparison with recent representative methods such as MoeQuant and EAQuant.

[1]: EAQuant: Enhancing Post-Training Quantization for MoE Models via Expert-Aware Optimization

[2]: MoEQuant: Enhancing Quantization for Mixture-of-Experts Large Language Models via Expert-Balanced Sampling and Affinity Guidance

**Questions:**

See weeknesses

---

> ### Author Response · Authors · 2025-11-21
> **Response to Reviewer (Part1)**
>
> We appreciate the reviewer’s recognition that our method offers a practical solution to MoE-LLM deployment bottlenecks, supported by clearly organized components, thorough analysis, and comprehensive evaluations. Below, we provide detailed point-by-point responses, with all relevant revisions incorporated into the manuscript.
>
>
> > W1: Insufficient theoretical support: Although the method is based on linear programming (LP), it does not provide any form of theoretical analysis or upper bound on the error.
>
> Thank you for the constructive comment.
> We would like to clarify that both the error approximation and the bit allocation in GEMQ are theoretically grounded. For error approximation, we follow the standard practice in Hessian-based quantization methods [1–3] and use a second-order local approximation (Eq. 3 and Eq. 4) to analyze the loss increase. Following [2], we further derive a computationally tractable estimate of this loss increase, which we use as the expert importance proxy in Eq. 6. For bit allocation, the Integer Linear Programming (ILP) formulation determines the optimal bit assignment under a global budget using the derived expert importance. It is worth noting that this ILP formulation is a variant of the multiple-choice knapsack problem (MCKP). A feasible solution is guaranteed as long as the target bit budget exceeds the cost of the minimal valid configuration (i.e., assigning all experts to the 1-bit). Consequently, the upper bound of the approximation error is strictly limited to the error of this minimal configuration.
>
>
> [1] Nagel, Markus, et al. "Up or Down? Adaptive Rounding for Post-Training Quantization." International conference on machine learning. PMLR, 2020.
>
> [2] Li, Yuhang, et al. "BRECQ: Pushing the Limit of Post-Training Quantization by Block Reconstruction." arXiv preprint arXiv:2102.05426 (2021).
>
> [3] Dong, Zhen, et al. "HAWQ-V2: Hessian Aware trace-Weighted Quantization of Neural Networks." Advances in neural information processing systems 33 (2020): 18518-18529.
>
>
> > W2: Lack of quantitative ablation in incremental quantization: The authors visualize the smoothness of the loss surface in the appendix, but do not provide quantitative indicators to explain the contribution or error accumulation at each stage.
>
> We apologize for the confusion. We would like to clarify that our method does **not** incur quantization error accumulation. At each stage, we compute the model statistics (i.e., the gradients and perturbations described in Eq. 6) using the quantized and fine-tuned model from the previous stage **solely for the purpose of bit allocation**. The actual quantization is always applied to the full-precision model rather than to an already-quantized model.
>
> We provide additional empirical results in the table below demonstrating the effectiveness of progressive quantization: computing model statistics from a quantized and fine-tuned model consistently outperforms using full-precision statistics, particularly in the low-bit regime. This finding is consistent with our theoretical analysis in Section 4.3.
>
> **Table: Ablation of Progressive Quantization (PQ) on Mixtral-8×7B**
> | BPE | Method | WT2$^\downarrow$ | C4$^\downarrow$ | 0-shot Avg$^\uparrow$ |
> |-----|---------|--------|--------|------------------|
> | 2.5 | w/o PQ (from FP16) | 5.04 | 9.02 | 64.53 |
> |     | **PQ (from 3.0-bpe)** | **5.03** | **9.01** | **64.82** |
> | 2.0 | w/o PQ (from FP16) | 6.17 | 11.14 | 59.11 |
> |     | **PQ (from 2.5-bpe)** | **6.09** | **11.00** | **59.60** |
> | 1.5 | w/o PQ (from FP16) | 9.66 | 23.53 | 48.24 |
> |     | **PQ (from 2.0-bpe)** | **9.01** | **21.88** | **49.87** |
>
>
> **Table: Ablation of Progressive Quantization (PQ) on DeepSeekV2-Lite**
> | BPE | Method | WT2$^\downarrow$ | C4$^\downarrow$ | 0-shot Avg$^\uparrow$ |
> |-----|---------|--------|--------|------------------|
> | 2.5 | w/o PQ (from FP16) | 6.86 | 10.54 | 60.48 |
> |     | **PQ (from 3-bpe)** | **6.83** | **10.50** | **60.59** |
> | 2.0 | w/o PQ (from FP16) | 7.75 | 12.20 | 54.00 |
> |     | **PQ (from 2.5-bpe)** | **7.74** | **12.12** | **54.32** |
> | 1.5 | w/o PQ (from FP16) | 11.72 | 21.49 | 45.17 |
> |     | **PQ (from 2.0-bpe)** | **11.30** | **20.37** | **45.51** |

---

> ### Author Response · Authors · 2025-11-21
> **Response to Reviewer (Part2)**
>
> > W3: Insufficient comparison with state-of-the-art methods: Only PMQ and Uniform are compared, without comparison with recent representative methods such as MoeQuant and EAQuant.
>
> We appreciate the additional related works. Below, we provide a brief discussion of the two methods together with additional experimental comparisons.
>
> ***Comparison with EAQuant.***
> EAQuant primarily focuses on **outlier suppression** under **uniform** bit-width **weight–activation** quantization, whereas our method derives an optimal **mixed-precision** strategy for **weight-only** quantization. Although EAQuant also examines router distribution shift, it relies on a rigid layer-wise alignment scheme that brings only marginal gains (e.g., <0.1 perplexity improvement reported). In contrast, our global router fine-tuning yields substantial improvements for quantized models, as shown in *Figure 5*, with a detailed comparison provided in *Table 3* and *Figure 8*. Moreover, EAQuant focuses on higher-bit settings (≥3 bpe), whereas our method targets more aggressive low-bit regimes (≤2.5 bpe) to better reduce the memory footprint of MoE expert weights.
>
> Below, we provide a comparison between EAQuant and the proposed GEMQ for Mixtral-8×7B quantization. As there is no shared quantization configuration between EAQuant and our method, we compare our **2.5-bpe model (W2.5A16)** and **3.0-bpe model (W3A16)** with the closest available setting in EAQuant, namely **W3A4**.
> Since there are slight differences between the FP16 baselines reported in the original paper and our own measurements, we additionally report the percentage increase after quantization relative to the FP16 baseline, computed as: $\Delta= (\\rm{Quantized}-\rm{FP16}) / \rm{FP16}$.
> As shown in the table, the 3-bpe mixed-precision GEMQ consistently outperforms EAQuant, while the 2.5-bpe model is on par with it.
>
>
>
> **Table: Comparison between EAQuant and GEMQ on Mixtral-8×7B**
>
> (*) denotes results from the original paper.
>
> | Method | WT2$^\downarrow$ | C4$^\downarrow$ | PIQA$^\uparrow$ | ARC-E$^\uparrow$ | ARC-C$^\uparrow$ | BoolQ$^\uparrow$ | Winogrande$^\uparrow$ | Avg$^\uparrow$ |
> |--------|--------|--------|--------|---------|---------|-----------|----------------|----------|
> | FP16* | 3.84 | 6.98 | 83.41 | 83.29 | 55.80 | 84.56 | 75.85 | 76.58 |
> | **EAQuant-W3A4*** | 5.27 | 8.23 | 79.05 | 78.45 | 50.68 | 78.69 | 69.30 | 71.23 |
> | (Δ%) | (+37.24) | (+17.91) | (-5.23) | (-5.81) | (-9.18) | (-6.94) | (-8.64) | (-6.98) |
> | FP16 | 3.84 | 7.40 | 83.68 | 83.42 | 59.73 | 85.05 | 76.32 | 77.64 |
> | **GEMQ-W3A16** | 4.37 | 8.06 | 82.54 | 80.68 | 57.34 | 85.02 | 74.90 | 76.10 |
> | (Δ%) | (**+13.78**) | (**+8.97**) | (**-1.36**) | (**-3.28**) | (**-4.00**) | (**-0.04**) | (**-1.86**) | (**-1.99**) |
> | **GEMQ-W2.5A16** | 5.03 | 9.02 | 81.07 | 75.93 | 51.79 | 80.09 | 73.72 | 72.52 |
> | (Δ%) | (+30.99) | (+21.89) | (-3.12) | (-8.98) | (-13.29) | (-5.83) | (-3.41) | (-6.59) |
>
>
> ***Comparison with MoEQuant.***
> MoEQuant focuses on constructing **optimal calibration data** for **uniform weight-only** MoE quantization via a self-sampling strategy. It also extends GPTQ with affinity-guided weighting to prioritize high-affinity tokens and reduce quantization error. However, it does **not** explore mixed-precision bit allocation or address the router distortion introduced by expert quantization. Similar to EAQuant, MoEQuant also operates in higher-bit regimes (≥3 bpe).
>
> Below, we compare MoEQuant and the proposed GEMQ for Mixtral-8×7B quantization. We compare our **2.5-bpe (W2.5A16)** and **3.0-bpe (W3A16)** models with the **W3A16** setting in MoEQuant. We also report the percentage increase relative to the FP16 baseline: $\Delta= (\\rm{Quantized}-\rm{FP16}) / \rm{FP16}$. The results in the table show that 3-bpe mixed-precision GEMQ consistently outperforms MoEQuant across all benchmarks.
>
>
> **Table: Comparison between MoEQuant and GEMQ on Mixtral-8×7B**
>
> (*) denotes results from the original paper.
>
> | Method | WT2$^\downarrow$ | C4$^\downarrow$ | BoolQ$^\uparrow$ | MathQA$^\uparrow$ | MMLU$^\uparrow$ | GSM8K$^\uparrow$ | Avg$^\uparrow$ |
> |--------|--------|--------|-----------|------------|------------|-------------|-----------|
> | FP16* | 3.84 | 6.87 | 85.23 | 42.41 | 70.50 | 65.88 | 66.01 |
> | **MoEQuant-W3A16*** | 4.90 | 8.24 | 82.81 | 38.82 | 64.10 | 43.21 | 57.24 |
> | (Δ%) | (+27.60) | (+19.94) | (-2.84) | (-8.46) | (-9.08) | (-34.41) | (-13.29) |
> | FP16 | 3.84 | 7.40 | 85.05 | 41.78 | 70.97 | 57.77 | 63.89 |
> | **GEMQ-W3A16** | 4.37 | 8.06 | 85.02 | 38.63 | 64.63 | 49.66 | 59.49 |
> | (Δ%) | (**+13.78**) | (**+8.97**) | (**-0.04**) | (**-7.54**) | (**-8.93**) | (**-14.04**) | (**-6.90**) |
> | **GEMQ-W2.5A16** | 5.03 | 9.02 | 80.09 | 34.10 | 59.74 | 42.30 | 54.06 |
> | (Δ%) | (+30.99) | (+21.89) | (-5.83) | (-18.38) | (-15.82) | (-26.78) | (-15.39) |

---

> ### Author Response · Authors · 2025-11-28
> **A Kind Reminder Regarding Our Responses to Your Comments**
>
> Dear Reviewer mxSw,
>
> Thank you once again for your time and valuable comments. As the discussion phase is approaching its end, we would like to kindly check whether our responses have adequately addressed your concerns. If there are any remaining questions or points that would benefit from further clarification, please feel free to let us know.
>
> We sincerely look forward to your feedback.
>
> Best regards,
>
> Authors of Paper 7529

---

### Author Response · Authors · 2025-12-02
**Summary of Discussion**

Dear Area Chair,

We sincerely appreciate the time and thorough consideration that you and the reviewers have devoted to evaluating our submission. Below, we provide a concise summary of the efforts made during the rebuttal to address the reviewers’ core concerns. All updates in the manuscript are highlighted in blue.

* **Concern: Insufficient baseline comparisons** (Reviewers **mxSw**, **Hrwz**)

  **Response**: We added quantitative comparisons with EAQuant and MoEQuant (SOTA MoE quantization methods), showing GEMQ consistently outperforms both in perplexity and zero-shot benchmarks on Mixtral-8x7B (Appendix A.3). We also compared against SpQR (a mixed-precision quantization method for dense LLMs), demonstrating that SpQR collapses in low-bit regimes while GEMQ remains robust (Appendix A.3).

* **Concern: Robustness and stability with small calibration set** (Reviewers **rBrH**, **Hrwz**)

  **Response**: We verified that expert importance estimates derived from 128 sequences align closely with those from 2048 sequences, demonstrating strong statistical stability (Section 5.2 & Appendix A.5). Robustness tests across multiple random seeds showed negligible variance in final model performance (Section 5.2).

* **Concern: Generalization and calibration bias** (Reviewers **rBrH**, **wRrM**, **Hrwz**)

  **Response**: We demonstrated that our method outperforms baselines on text modeling and downstream linguistic benchmarks under the same generic calibration data (Section 5.1). We further showed that incorporating a small amount of math calibration data significantly improves performance on both linguistic benchmarks and challenging math reasoning tasks, again surpassing the baselines under the same calibration data (Section 5.1 & Appendix A.2). We also discussed constructing more representative calibration datasets as an important direction for future work (Appendix A.2).

* **Concern: Theoretical justifications of global expert importance** (Reviewers **mxSw**, **Hrwz**)

  **Response**: We provided further explanation on the theoretical grounding behind the global expert bit allocation in our response to reviewer **mxSw**.
  We also clarified the misunderstanding from reviewer **Hrwz** regarding our importance proxy and justified its design choices (e.g., using layer output gradients and second-order approximation) in our response to reviewer **Hrwz**.

* **Concern: Efficiency and necessity of router fine-tuning** (Reviewers **wRrM**, **Hrwz**)

  **Response**: We clarified the misunderstanding raised by reviewer **wRrM** and **Hrwz** regarding our router fine-tuning (RFT) strategy by demonstrating its lightweight yet critical nature. We also provided a detailed and intuitive explanation of RFT in our responses to both reviewers and in Section 4.2.

* **Concern: Impact of progressive quantization** (Reviewers **mxSw**, **wRrM**)

  **Response**: We added quantitative ablation results showing that progressive quantization consistently outperforms the baseline without it, which becomes especially critical when operating below 2 bits (Appendix A.7).

* **Concern: Missing ablations on additional LP constraints** (Reviewers **wRrM**, **Hrwz**)

  **Response**: We added ablations on various constraint settings and validated that the constraint "at least one higher-bit expert per layer" acts as a necessary regularizer for the LP solver in ultra-low-bit settings, preventing degenerate solutions when the importance proxy becomes less accurate (Section 4.1 & Appendix A.5).

* **Concern: Limited exploration of bit-width choices** (Reviewer **rBrH**)

  **Response**: We expanded the expert bit-width candidate set from {1,2,3} to {0,1,2,3,4} and evaluated multiple attention bit-width settings, demonstrating the flexibility of our global LP formulation (Appendix A.5).

We believe that the quality of our revised manuscript, together with the additional experiments and clarifications, sufficiently addresses the concerns raised.

Sincerely,

Authors of Submission 7529

---

### Meta-Review · Area_Chair_HrHD · 2025-12-20

**Summary:**

**Paper summary.** This paper proposes GEMQ, a mixed-precision quantization pipeline for MoE LLMs. The two main ideas are: (1) allocate expert bit-widths *globally* across the whole model using an LP formulation (instead of per-layer heuristics), and (2) do light router fine-tuning after quantization so routing adapts to quantized experts. The paper also uses a progressive quantization schedule. The goal is to reduce memory/compute with minimal quality loss.

**What happened in the discussion.** Reviewers largely agreed the topic is important, but asked hard questions about (a) how expert “importance” is estimated (especially with small calibration sets and dataset bias), (b) why the LP needs extra constraints like “each layer must keep some higher-bit experts”, (c) how robust the pipeline is across tasks and calibration choices, and (d) whether the explored bit-width design space is too narrow. In the forum, the authors added several missing pieces: new baseline comparisons (EAQuant and MoEQuant on Mixtral-8x7B; also SpQR as a dense-model mixed-precision baseline), stability checks for importance estimates using 128 vs 2048 sequences and multiple seeds, ablations for different LP constraint settings, and an expanded candidate bit set (including {0,1,2,3,4}) with more attention-precision variants. They also wrote a concise “Summary of Discussion” to tie these points together.

**My assessment as AC.** After reading the paper and the discussion, I think this work is *close* to the bar: the pipeline is coherent, and the added experiments address several reviewer concerns. That said, even in the updated form, the core novelty is moderate (global LP allocation + router tuning), and the story still depends on a sequence of design choices that are not strongly motivated from first principles (importance proxy choice, constraint design, calibration data choices). In a less competitive year, I could imagine this being accepted as a solid systems paper. In this batch, I do not think it is among the strongest ~30%.

**Decision.** Reject (borderline). This is not a statement that the paper is “bad”. It is a reasonable piece of work, but ICLR is very competitive and different reviewers/ACs weigh novelty vs. engineering evidence differently. I encourage the authors to keep the stronger baseline comparisons and the robustness/ablation results in the next version, and to simplify the narrative around the importance proxy and LP constraints.

**Reviewer Concerns:**

- **Reviewer rBrH (rating 6, confidence 5)**: Wanted a broader bit-width design space and robustness analysis (calibration noise, richer candidate sets, hardware-aware trade-offs). Authors expanded the candidate bit set and added more variants/ablations. **Status:** partially resolved (still limited hardware-aware analysis).
- **Reviewer Hrwz (rating 4, confidence 5)**: Raised repeated concerns about the importance proxy and its robustness with small calibration data (e.g., 128 sequences), dataset shift of expert activation, and the need/meaning of LP constraints. Authors added stability experiments (128 vs 2048 sequences), explained why they use output gradients rather than expert-weight gradients, and added constraint ablations. Reviewer still posted follow-up questions after the first response. **Status:** partially resolved; some skepticism remains.
- **Reviewer wRrM (rating 4, confidence 4)**: Focused on sensitivity/tuning interactions between LP allocation and router fine-tuning, and asked about the necessity and effect of progressive quantization and constraints. Authors added quantitative ablations for progressive quantization and constraint settings and clarified router fine-tuning as lightweight but important. **Status:** mostly resolved.
- **Reviewer mxSw (rating 4, confidence 3)**: Asked for stronger theoretical justification and more ablations; authors added more explanation and ablations, and expanded baseline comparisons. **Status:** partially resolved.

**Reviewer Scores:**

- **rBrH (6,5)**: Likely unchanged.
- **Hrwz (4,5)**: Likely unchanged (still skeptical even after detailed replies).
- **wRrM (4,4)**: Could increase slightly after ablations, but no explicit score update is recorded.
- **mxSw (4,3)**: Likely unchanged.

---

### Decision · Program_Chairs · 2026-01-26

Reject